# Bidirectional crosstalk between Hypoxia-Inducible Factor and glucocorticoid signalling in zebrafish larvae

**Davide Marchi**[1]*, **Kirankumar Santhakumar**[2], **Eleanor Markham**[1], **Nan Li**[3], **Karl-Heinz Storbeck**[4], **Nils Krone**[3,5], **Vincent T. Cunliffe**[1], **Fredericus J. M. van Eeden**[1]*

**1** The Bateson Centre & Department of Biomedical Science, Firth Court, University of Sheffield, Western Bank, Sheffield, United Kingdom, **2** Department of Genetic Engineering, SRM Institute of Science and Technology Kattankulathur, India, **3** The Bateson Centre & Department of Oncology and Metabolism, School of Medicine, University of Sheffield, Sheffield, United Kingdom, **4** Department of Biochemistry, Stellenbosch University, Stellenbosch, Matieland, South Africa, **5** Department of Medicine III, University Hospital Carl Gustav Carus, Technische Universität Dresden, Dresden, Germany

* dmarchi1@sheffield.ac.uk (DM); f.j.vaneeden@sheffield.ac.uk (FJMv)

**Data Availability Statement:** All relevant data are within the manuscript and its Supporting Information files.

## Abstract

In the last decades *in vitro* studies highlighted the potential for crosstalk between Hypoxia-Inducible Factor-(HIF) and glucocorticoid-(GC) signalling pathways. However, how this interplay precisely occurs *in vivo* is still debated. Here, we use zebrafish larvae (*Danio rerio*) to elucidate how and to what degree hypoxic signalling affects the endogenous glucocorticoid pathway and vice versa, *in vivo*. Firstly, our results demonstrate that in the presence of upregulated HIF signalling, both glucocorticoid receptor (Gr) responsiveness and endogenous cortisol levels are repressed in 5 days post fertilisation larvae. In addition, despite HIF activity being low at normoxia, our data show that it already impedes both glucocorticoid activity and levels. Secondly, we further analysed the *in vivo* contribution of glucocorticoids to HIF activity. Interestingly, our results show that both glucocorticoid receptor (GR) and mineralocorticoid receptor (MR) play a key role in enhancing it. Finally, we found indications that glucocorticoids promote HIF signalling via multiple routes. Cumulatively, our findings allowed us to suggest a model for how this crosstalk occurs *in vivo*.

## Author summary

Hypoxia is a common pathophysiological condition to which cells must rapidly respond in order to prevent metabolic shutdown and subsequent death. This is achieved via the activity of Hypoxia-Inducible Factors (HIFs), which are key oxygen sensors that mediate the ability of the cell to cope with decreased oxygen levels. Although it aims to restore tissue oxygenation and perfusion, it can sometimes be maladaptive and contributes to a variety of pathological conditions including inflammation, tissue ischemia, stroke and growth of solid tumours. In this regard, synthetic glucocorticoids which are analogous to naturally occurring steroid hormones, have been used for decades as anti-inflammatory drugs for treating pathological conditions which are linked to hypoxia (i.e. asthma, rheumatoid

**Funding:** DM was awarded by a University of Sheffield, Biomedical Science department studentship (https://www.sheffield.ac.uk/) and supported by two BBRSC grants to FVE (BB/R015457/1; BB/M02332X/1). FVE was funded by two Biotechnology and Biological Sciences Research Council (BBRSC: https://bbsrc.ukri.org/) grants (code 1: BB/R015457/1; code 2: BB/M02332X/1). NK was funded by the Deutsche Forschungs Gesellschaft (https://www.dfg.de/) grant (KR3363/3-1). The funders had no role in study design, data collection and analysis, decision to publish, or preparation of the manuscript.

**Competing interests:** The authors have declared that no competing interests exist.

arthritis, ischemic injury). Indeed, previous *in vitro* studies highlighted the presence of a crosstalk between HIF and glucocorticoids. However, how this interplay precisely occurs in an organism and what the molecular mechanism is behind it are questions that still remain unanswered. Here, we provide a thorough *in vivo* genetic analysis, which allowed us to propose a logical model of interaction between glucocorticoid and HIF signalling. In addition, our results are important because they suggest a new route to downregulate HIF for clinical purposes.

## Introduction

Glucocorticoids (GC) constitute a well-characterized class of lipophilic steroid hormones produced by the adrenal glands in humans and by the interrenal tissue in teleosts. The circadian production of glucocorticoids in teleosts is regulated by the hypothalamus-pituitary-interrenal (HPI) axis, which is the equivalent of the mammalian hypothalamus-pituitary-adrenal (HPA) axis. Both are central to stress adaptation [1–4]. Interestingly, both in humans and teleosts cortisol is the primary glucocorticoid and regulates a plethora of physiological processes including glucose homeostasis, inflammation, intermediary metabolism and stress response [5]. In particular, cortisol can exert these functions via direct binding both to the glucocorticoid receptor (Gr) and to the mineralocorticoid receptor (Mr), which bind cortisol with different affinities. [3,6]. Together they act as a transcription factor, which can function either in a genomic or in non-genomic way [5,7–9].

Hypoxia-inducible factor (HIF) transcription factors are key regulators of the cellular response to hypoxia, which coordinate a metabolic shift from aerobic to anaerobic metabolism in the presence of low oxygen availability in order to assure homeostasis [10]. In mammals there are at least three isoforms of HIF-α (HIF-1α, HIF-2α and HIF-3α) and two main isoforms of HIF-1β (ARNT1 and ARNT2). [11]. Interestingly, due to a genome duplication event, there are two paralogs for each of the three Hif-α isoforms (Hif-1αa, Hif-1αb, Hif-2αa, Hif-2αb, Hif-3αa and Hif-3αb) in zebrafish. Among these, Hif-1αb is thought to be the key zebrafish homologue in the hypoxic response [12]. With respect to *HIF-1β (ARNT)* paralogues, the expression of two genes encoding Arnt1 and Arnt2 proteins has been described in zebrafish [13–16].

Whilst ARNT is constitutively expressed, the cytoplasmic HIF-α subunits are primarily regulated post-translationally via the PHD3-VHL-E3-ubiquitin ligase protein degradation complex. This is believed to occur in order to allow a rapid response to decreasing oxygen levels [12,17–19]. Indeed, hypoxia, is a common pathophysiological condition [20,21] to which cells must promptly respond in order to avert metabolic shutdown and subsequent death [12]. In the presence of normal oxygen levels, a set of prolyl hydroxylases (PHD1, 2 and 3) use the available molecular oxygen directly to hydroxylate HIF-α subunit. Hydroxylated HIF-α is then recognised by the Von Hippel Lindau (VHL) protein, which acts as the substrate recognition part of a E3-ubiquitin ligase complex. This leads to HIF-α proteasomal degradation to avoid HIF pathway activation under normoxic conditions. On the other hand, low $O_2$ levels impair the activity of the PHD enzymes leading to HIF-α stabilisation and subsequent translocation in the nucleus. Here, together with the HIF-β subunit, HIF-α forms a functional transcription complex, which drives the hypoxic response [22]. Although the HIF response is aimed to restore tissue oxygenation and perfusion, it can sometimes be maladaptive and can contribute to a variety of pathological conditions including inflammation, tissue ischemia, stroke and growth of solid tumours [23]. Finally, it is important to note for this study that HIF signalling

is able to regulate its own activation via negative feedback, by inducing the expression of PHD genes, in particular prolyl hydroxylase 3 (PHD3) [24,25].

The presence of a crosstalk between glucocorticoids and hypoxia dependent signalling pathways has been reported in several *in vitro* studies [26–30]. Moreover, synthetic glucocorticoids (ie. betamethasone and dexamethasone), which are analogous to naturally occurring steroid hormones, have been extensively used for decades as anti-inflammatory drugs for treating pathological conditions which are linked to hypoxia (i.e. asthma, rheumatoid arthritis, ischemic injury, etc.) [31–33]. However, due to the presence of adverse effects [34] and glucocorticoid resistance [35,36], their use has been limited. Therefore, extending the research on how precisely this interplay occurs *in vivo*, may have a wide physiological significance in health and disease.

The first evidence of interaction between HIF and GR was provided by Kodama et al. 2003 [26], who discovered that ligand-dependent activation of glucocorticoid receptor enhances hypoxia-dependent gene expression and hypoxia response element (HRE) activity in HeLa cells. Leonard et al. 2005 [27] subsequently revealed that GR is transcriptionally upregulated by hypoxia in human renal proximal tubular epithelial cells. Furthermore, the hypoxic upregulation of GR was confirmed by Zhang et al 2015 [29]. In contrast, a dexamethasone-mediated inhibition of HIF-1α target genes expression in hypoxic HEPG2 cells was demonstrated by Wagner et al. 2008 [28]. In addition to that, they showed retention of HIF-1α in the cytoplasm, suggesting a blockage in nuclear import. Finally, Gaber *et al.*, 2011 [37] indicated the presence of dexamethasone-induced suppression of HIF-1α protein expression, which resulted in reduced HIF-1 target gene expression.

From these *in vitro* results it has become clear that the HIF-GC crosstalk is complex and may depend on cell type. In the present study, we have used the zebrafish (*Danio rerio*) as an *in vivo* model organism to study how and to what degree hypoxic signalling affects the endogenous glucocorticoids' response and vice versa. The use of whole animals allows us to show how these signals interact at a more global level than in cell culture, where interactions between different tissues and cell types are not easily modelled. The zebrafish offers an excellent genetic vertebrate model system for endocrine studies, and similar to humans, they are diurnal and use cortisol as the main glucocorticoid hormone [38]. Importantly, unlike other teleosts, zebrafish have only a single glucocorticoid (zGr) and mineralocorticoid receptor (Mr) (zMr) isoform [3]. Moreover, zGr shares high structural and functional similarities to its human equivalent, making zebrafish a reliable model for studying glucocorticoids activity *in vivo* [39–41]. Additionally, zebrafish share all the components of the human HIF signalling pathway and it has been proved to be a very informative and genetically tractable organism for studying hypoxia and HIF pathway both in physiological and pathophysiological conditions [12,25,42].

In our previous work, we identified new activators of the HIF pathway, e.g. betamethasone, a synthetic glucocorticoid receptor agonist [43]. Counterintuitively, GR loss of function was shown by Facchinello and colleagues to hamper the transcriptional activity linked to immune-response (i.e of cytokines Il1β, Il8 and Il6 and of the metalloproteinase Mmp-13) [5]. Finally, glucocorticoid receptor has been also found to synergistically activate proinflammatory genes by interacting with other signalling pathways [40,44–46].

In the present study, we utilised both a genetic and pharmacological approach to alter these two pathways during the first 120 hours post fertilisation of zebrafish embryos. In particular, we took advantage of two different mutant lines we have generated (*hif1β*$^{sh544}$ (*arnt1*) and *gr*$^{sh543}$ (*nr3c1*) respectively), coupled to an already existing *vhl*$^{hu2117/+}$;*phd3:eGFP*$^{i144/i144}$ hypoxia reporter line [25], to study the effect of HIF activity on GC signalling and vice-versa, via a "gain-of-function/loss-of-function" approach. Phenotypic and molecular analyses of these mutants have been accompanied by optical and fluorescence microscope imaging.

Importantly, we not only confirm that betamethasone is able to increase the expression of *phd3:eGFP*, a marker of HIF activation in our zebrafish HIF-reporter line, but we also show that BME-driven HIF response requires Hif1β/Arnt1 action to occur.

Furthermore, our results also demonstrate that both Gr and Mr loss of function are able to partially rescue *vhl* phenotype, allowing us to confirm the importance of glucocorticoids in assuring high HIF signalling levels. This finding may have wider significance in health and disease, as so far it is proven difficult to downregulate HIF signalling.

Our results also demonstrate that in the presence of upregulated HIF pathway (by mutating *vhl*), both the glucocorticoid receptor activity and the endogenous cortisol levels are repressed in 5 dpf larvae, whereas when the HIF pathway is suppressed (by mutating *hif1β*) they are significantly increased. Finally, qPCR analysis on GC target genes, *in situ* hybridisation on the expression of steroidogenic genes and cortisol quantification on the aforementioned mutant lines confirmed our hypothesis.

Taken together, these results allow us to deepen the knowledge of how the crosstalk between HIF and glucocorticoid pathway occurs *in vivo* and to underscore a new model of interaction between these two major signalling pathways.

## Results

### Generating *arnt1* and *arnt1;vhl* knockout in zebrafish

To study the interplay between HIF and GC signalling *in vivo*, using a genetic approach, we required a Hif1β/Arnt1 mutant line (in a *phd3:eGFP;vhl*$^{+/-}$ background) to enable the downregulation of the HIF pathway. Hif-1β (hypoxia-inducible factor 1 beta, Arnt1) is a basic helix-loop-helix-PAS protein which translocates from the cytosol to the nucleus after ligand binding to Hif-α subunits, after the stabilization of the latter in the cytoplasm. It represents the most downstream protein in the HIF pathway and for this reason it is the most suitable target.

Using CRISPR/Cas9 mutagenesis we obtained a 7 bp insertion in exon 5 (coding bHLH DNA binding domain (DBD) of the Hif-1β protein; allele name sh544) in *vhl* heterozygote embryos (**Fig 1A**). The resulting frameshift mutation was predicted to lead to a premature stop codon at the level of the DNA-binding domain, which would result in a severely truncated protein. The resulting line *hif1β*$^{sh544/+}$*;vhl*$^{hu2117/+}$*;phd3:eGFP*$^{i144/i144}$ will be called *arnt1*$^{+/-}$*; vhl*$^{+/-}$, whereas the *vhl*$^{hu2117/+}$*;phd3:eGFP*$^{i144/i144}$ line will be called *vhl*$^{+/-}$ hereafter.

Initial analysis performed on *arnt1*$^{+/-}$*;vhl*$^{+/-}$ incross-derived 5 dpf larvae (F1 generation) confirmed the suppressive effect that *arnt1* mutation was expected to have on *vhl* mutants. Overall, *arnt1*$^{-/-}$*;vhl*$^{-/-}$ larvae showed a substantially attenuated *vhl* phenotype, characterized by a reduced *phd3:eGFP* related brightness, especially in the liver (**Fig 1C'**), with the absence of pericardial edema, excessive caudal vasculature and normal yolk usage (**Fig 1C**) compared to *vhl*$^{-/-}$ larvae (**Fig 1B and 1B'**). In particular, this was quantified as a 39% downregulation (P<0.0017) at the level of the head, a 75% downregulation (P<0.0001) in liver and a 58% downregulation (P<0.0001) in the rest of the body (from the anus to the caudal peduncle), in terms of *phd3: eGFP*-related brightness, compared to *vhl*$^{-/-}$ larvae (**Figs 1C'**, **1B'**, **S1A and S1D**).

Furthermore, since homozygous *vhl* mutants are lethal by 8-10dpf [47], we analysed the efficacy of *arnt1* mutation in rescuing *vhl* phenotype. To this end, we attempted to raise *arnt1*$^{-/-}$*;vhl*$^{-/-}$ after day 5 post fertilization. Notably, double mutants were able to survive beyond 15 dpf, but failed to grow and thrive when compared to their wild-type siblings, which led us to euthanise them due to health concerns at 26 dpf (**S1B Fig**). Of note, *arnt1* homozygotes, in a *vhl*$^{/+}$ or wt background, were morphologically indistinct and adults were viable and fertile. In contrast, the previously published *arnt2*$^{-/-}$ zebrafish larvae were embryonic lethal around 216 hpf [14].

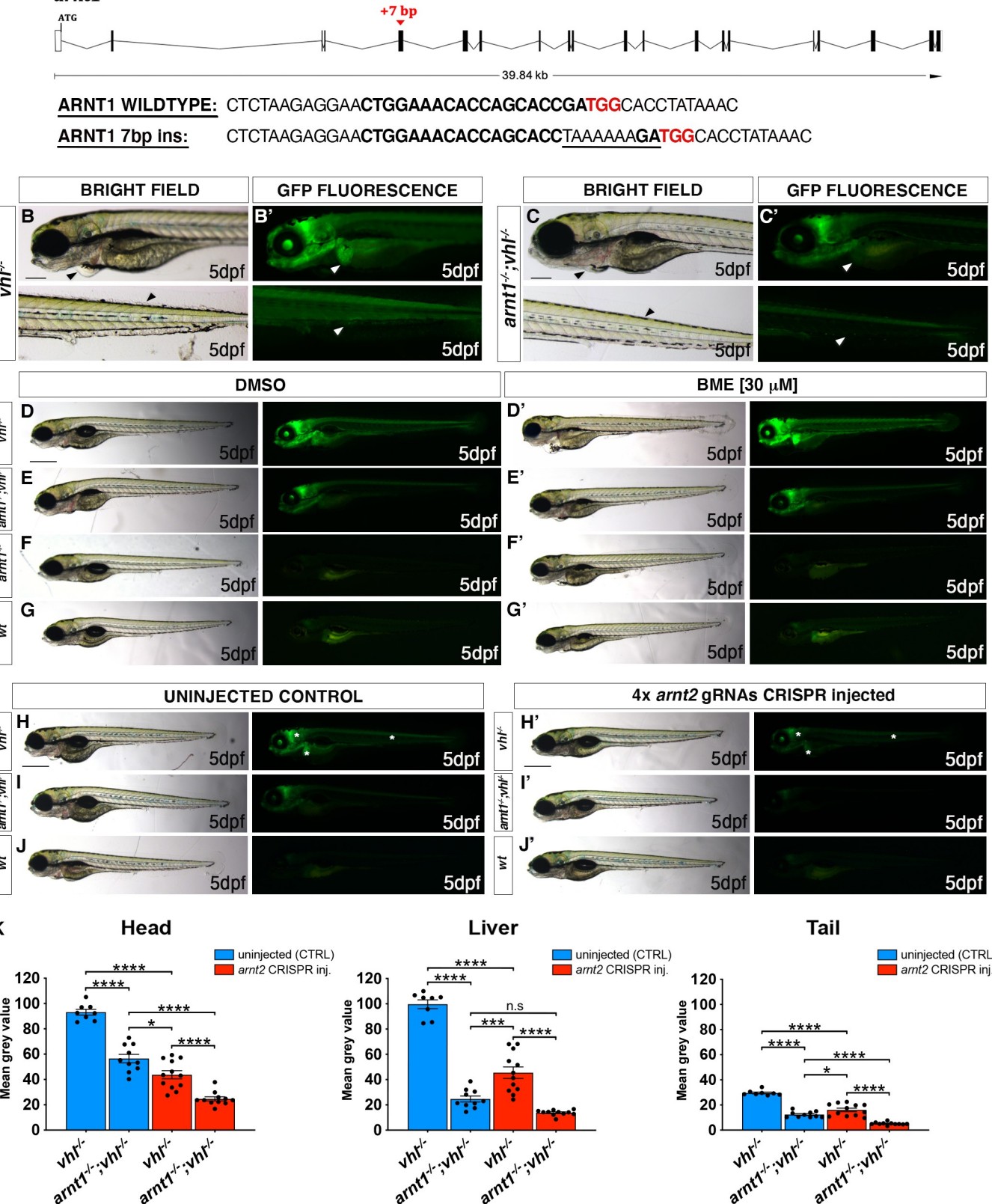

**A** *arnt1*

ARNT1 WILDTYPE: CTCTAAGAGGAA**CTGGAAACACCAGCACCGA**TGGCACCTATAAAC

ARNT1 7bp ins: CTCTAAGAGGAA**CTGGAAACACCAGCACC**TAAAAAAGATGGCACCTATAAAC

**Fig 1. *arnt1* and *arnt2* have partially overlapping functions and synergistically contribute to HIF signalling.** A. Schematic representation of zebrafish *hif1β* (*arnt1*) gene. Exons are shown as black boxes, whereas introns as lines. The red arrowhead shows the position of a +7 bp insertion in exon 5 (encoding the bHLH DNA binding domain). In the *arnt1* wt and mutant sequence. CRISPR target site: bold. Protospacer-adjacent-motif (PAM) sequence: red. B-B'. Magnified picture of a representative 5 dpf *vhl*$^{-/-}$ larva compared to 5dpf *arnt1*$^{-/-}$;*vhl*$^{-/-}$ (C-C'). Among the 120 GFP$^+$ embryos derived from *arnt1*$^{+/-}$;*vhl*$^{+/-}$(*phd3: eGFP*) x *arnt1*$^{-/-}$;*vhl*$^{+/-}$(*phd3:eGFP*), 15 larvae were characterized by the absence of pericardial oedema, no ectopic extra vasculature at the level of the tail, no bright liver and a reduced brightness in the rest of the body (black and white arrowheads). Genotyping post phenotypic analysis on sorted larvae confirmed genotype-phenotype correlation. Fluorescence, exposure = 2 seconds. Scale bar 200 μm. D-G. Representative picture of phenotypic analysis performed on DMSO and BME [30 μM] treated 5 dpf larvae, derived from *arnt1*$^{+/-}$;*vhl*$^{+/-}$(*phd3:eGFP*) x *arnt1*$^{-/-}$;*vhl*$^{+/-}$(*phd3:eGFP*) (n = 540). All the genotype combinations observed are represented in the figure. Among the 405 GFP$^+$ larvae, all the 25 *arnt1*$^{-/-}$;*vhl*$^{-/-}$ showed the aforementioned partially rescued vhl phenotype (D). Fluorescence, exposure = 2 seconds. Scale bar 500 μm. H-J. Representative pictures of 5 dpf CRISPANT mutants created by redundantly targeting *arnt2* gene via co-injection of 4x gRNAs in *arnt1*$^{+/-}$;*vhl*$^{+/-}$(*phd3:eGFP*) x *arnt1*$^{-/-}$; *vhl*$^{+/-}$(*phd3:eGFP*) derived embryos (n = 300). Uninjected embryos were used as control (n = 120). White asterisks: head, liver and tail regions. Fluorescence, exposure = 991,4 ms. Scale bar 500 μm. K. Statistical analysis performed on mean grey values quantification (at the level of the head, liver and tail), after phenotypic analysis on 5 dpf *arnt2* 4x gRNAs injected and uninjected larvae. *vhl*$^{-/-}$ uninjected n = 8 larvae: head 93.1 ± 2.33 (mean ± s.e.m); liver 99.65 ± 3.49 (mean ± s.e.m); tail 29.58 ± 0.73 (mean ± s.e.m). *arnt1*$^{-/-}$;*vhl*$^{-/-}$ uninjected n = 10 larvae: head 56.49 ± 3.36 (mean ± s.e.m); liver 24,7 ± 2.36 (mean ± s.e.m); tail 12.39 ± 0,75 (mean ± s.e.m). *vhl*$^{-/-}$ injected n = 12 larvae: head 43.69 ± 3.25 (mean ± s.e.m); liver 45.54 ± 4.57 (mean ± s.e.m); tail 16.09 ± 1.37 (mean ± s.e.m). *arnt1*$^{-/-}$;*vhl*$^{-/-}$ injected n = 11 larvae: head 24.66 ± 1.63 (mean ± s.e.m); liver 13.88 ± 0.66 (mean ± s.e.m); tail 5.16 ± 0.33 (mean ± s.e.m). Ordinary One-way ANOVA followed by Sidak's multiple comparison test (*P < 0.05; **P < 0.01; ***P <0.001; ****P < 0.0001).

## Arnt1 and Arnt2 are mutually required for HIF signalling in zebrafish

As *arnt1*;*vhl* double mutants still activate the *phd3:eGFP* HIF reporter, we examined the importance of Arnt2 isoform in the HIF pathway. Phenotypic analysis was carried out on 5 dpf Arnt2 CRISPANTs, created both in a *vhl*$^{+/-}$ and *arnt1*$^{+/-}$;*vhl*$^{+/-}$ background, according to the protocol of Wu et al., 2018 [48]. By analysing the expression of the *phd3:eGFP* transgene, we observed that *arnt2* CRISPR injected *vhl* mutants were characterized by a significant down-regulation of *phd3:eGFP*-related brightness at the level of the head (equals to 53%, P<0.0001), in the liver (equals to 54%, P<0.0001) and in the rest of the body (equals to 46%, P<0.0001), compared to uninjected *vhl* mutant larvae (**Fig 1H'** compared to **1H, white asterisks; Fig 1K**).

Furthermore, when both *arnt1* and *arnt2* isoforms were simultaneously knocked-out (**Fig 1I'**), the downregulation was even stronger at the level of the head (equals to 74%, P<0.0001), the liver (equals to 86%, P<0.0001) and in the rest of the body (equals to 83%, P<0.0001) (**Fig 1I'** compared to **1H; Fig 1K**). Of note, *phd3:eGFP*-related brightness in these mutants was still slightly higher than wildtype, (not shown; these levels are undetectable). Overall, these data show that Arnt1, even if not fundamental for survival, is the main isoform in the zebrafish liver required for HIF signalling, whereas Arnt2 is more expressed in the developing central nervous system (CNS), as reported by Hill et., al 2009. Of note, since both isoforms can form a functional complex with Hif-α isoforms and appear to function in the same organs, this allows us to confirm that they have partially overlapping functions *in vivo* and to show that they synergistically contribute to the HIF response.

## Modulation of HIF signalling affects GR signalling

To investigate the interaction between HIF and glucocorticoid signalling, we quantified the expression of four potential glucocorticoid target genes from mammalian studies (*fkbp5*, *il6st*, *pck1 and lipca*) both in a HIF upregulated (*vhl*$^{-/-}$), and downregulated scenario (*arnt1*$^{-/-}$) via RTqPCR analysis on 5 dpf larvae. We confirmed that in zebrafish larvae, *fkbp5* is the most sensitive and well-established readout of Gr activity [5,49,50], whilst the other aforementioned genes do not directly take part in the GC-GR negative feedback loop. Therefore, we focused this analysis on *fkbp5*.

Interestingly, our analysis shows that the expression of *fkbp5* is downregulated (fold change = 0.1; P = 0.0035) in the presence of an upregulated HIF pathway (*vhl*$^{-/-}$) compared to DMSO treated *vhl* siblings (**Fig 2A**). Vice versa, when the HIF pathway is suppressed

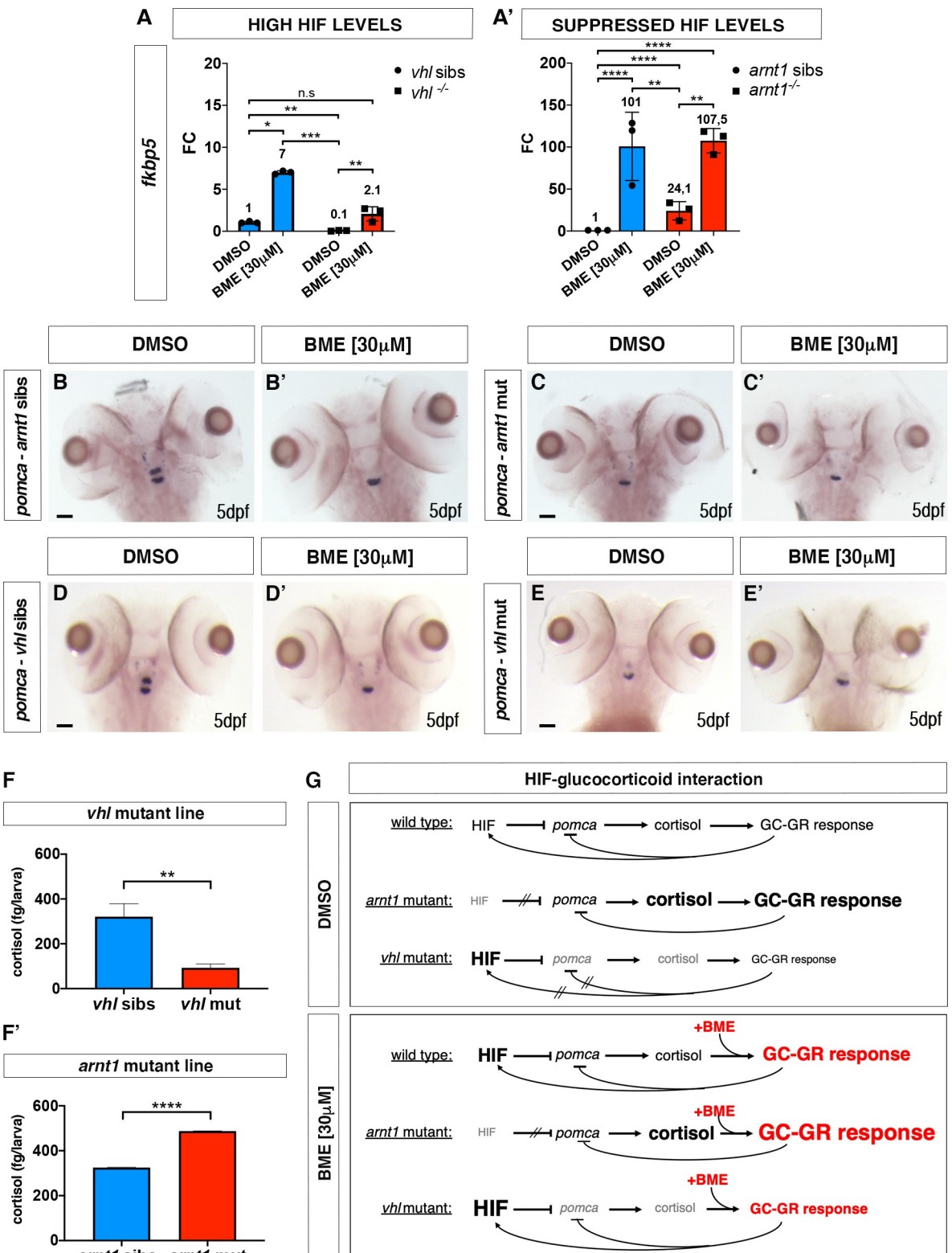

**Fig 2. HIF signalling inversely correlate with GC transcriptional activity and cortisol biosynthesis.** A. Schematic view of RTqPCR analysis on fkbp5 expression performed on the $vhl^{+/-}$ ($phd3:eGFP$) and the $arnt1^{+/-}$ ($phd3:eGFP$) mutant lines at 5 dpf:. Upregulated (in $vhl^{-/-}$) HIF signalling repressed Gr activity, whereas arnt1 loss of function derepressed it. Statistical analysis was performed on ΔΔCt values, whereas data are shown as fold change values for RTqPCR analysed samples; ordinary Two-way ANOVA followed by Dunnett's multiple

comparison test ($^{*}$P < 0.05; $^{**}$P < 0.01; $^{***}$P <0.001; $^{****}$P < 0.0001). B-C'. Representative pictures of WISH performed on DMSO and BME [30 µM] treated arnt1 mutant line, at 5 dpf, using pomca as probe. arnt1 wt DMSO treated (n = 30/30 larvae) showed normal pomca expression; arnt1 wt BME treated (n = 29/30 larvae) showed downregulated pomca expression. In contrast, arnt1$^{-/-}$ DMSO treated (n = 28/30) and arnt1$^{-/-}$ BME treated (n = 30/30) larvae showed downregulated pomca expression. Chi-square test ($^{****}$P < 0.0001). Scale bar 50 µm. D-E'. Representative pictures of WISH performed on DMSO and BME [30 µM] treated vhl mutant line, at 5 dpf, using pomca as probe. DMSO treated vhl siblings (n = 26/28) showed normal pomca expression; BME treated vhl siblings (n = 28/30) showed downregulated pomca expression. In contrast, vhl$^{-/-}$ DMSO (n = 28/29) and BME (n = 28/28) treated larvae showed downregulated pomca expression. Chi-square test ($^{****}$P < 0.0001). Scale bar 50 µm. F-F'. Steroid quantification results showed a significantly reduced cortisol concentration (P value <0;0028) in vhl mutants (92.7 fg/larva, in triplicate), compared to vhl siblings (321 fg/larva, in triplicate) at 5 dpf (F). Moreover, a significantly increased cortisol concentration (P value <0;0001) was measured in arnt1 mutants (487.5 fg/larva, in triplicate), compared to arnt1 wild-types (325 fg/larva, in triplicate) at 5 dpf (F'); unpaired t-test ($^{**}$P < 0.01; $^{***}$P <0.001). G. **DMSO**: Speculative scheme of how the putative HIF-GC crosstalk occurs in wildtypes and how it is affected both in arnt1$^{-/-}$ and in vhl$^{-/-}$ larvae at 5 dpf. In wildtype scenario HIF signalling helps the GC-GR negative feedback to protect the body from an uncontrolled stress response. In particular, we speculate that HIF transcriptional activity is able to inhibit pomca expression when cortisol levels arise over a certain threshold in order to maintain both HIF and GC basal levels. However, in arnt1$^{-/-}$ scenario, the HIF-mediated negative feedback is compromised by the lack of a functional Arnt1. This triggers an initial uncontrolled pomca expression, which increases cortisol levels and subsequently downregulate pomca expression itself. Vice versa, in vhl$^{-/-}$ scenario, the HIF-mediated negative feedback can exert a stronger inhibition of pomca due to the presence of upregulated HIF signalling. This results both in downregulated cortisol levels and in a suppressed GR responsiveness. However, the presence of alternative mechanisms cannot be completely excluded (i.e HIF might interact more directly with GC/GR to impair its function). Finally, the combination high cortisol/low pomca is very rare and this combination may change over the course of development. G. **BME**: Speculative scheme of how the putative HIF-GC crosstalk occurs in wildtypes and how it is affected both in arnt1$^{-/-}$ and in vhl$^{-/-}$ larvae at 5 dpf, after BME[30µM] treatment. In all the cases, because of betamethasone acts downstream of the HPI axis, by binding directly to Gr, it is able to upregulate glucocorticoid target genes expression. Consequently, since GC are able to stimulate HIF signalling, as expected, we observed an increase phd3:eGFP-related brightness both in wildtypes and in vhl$^{-/-}$. However, the fact that we did not observed any HIF upregulation both in arnt1$^{-/-}$ and in arnt1$^{-/-}$;vhl$^{-/-}$, highlighted the fact that the BME-induced HIF signalling activation is an Arnt1 dependent mechanism.

(arnt1$^{-/-}$), fkbp5 expression is upregulated (fold change = 24.1; P<0.0001), compared to DMSO treated wild-type levels (**Fig 2A'**).

To further examine the effect of HIF signalling on glucocorticoid responsiveness, we also performed betamethasone (BME) treatment [30 µM] on the aforementioned mutant lines, followed by RTqPCR analysis. Of note, BME was able to increase fkbp5 expression in vhl siblings and was only able to mildly do that in vhl mutants. Indeed, its induction levels appeared not only lower in BME treated vhl mutants (fold change = 2.1) than in BME treated siblings (fold change = 7, P = 0.0286), but also its expression was not significantly different from DMSO treated wild-types (**Fig 2A**). In contrast, when the HIF pathway was suppressed (arnt1$^{-/-}$), BME treatment was able to further upregulate the expression of fkbp5 (fold change = 107,5; P = 0.0031), compared to DMSO treated arnt1 mutants (**Fig 2A'**).

Collectively, we speculate that the upregulated HIF levels are able to repress the glucocorticoid receptor activity and can blunt its responsiveness to an exogenous GR agonist (BME treatment). On the other hand, importantly, although HIF activity is expected to be low in wild-type larvae in a normoxic environment, its function is also detectable with respect to suppression of GR activity. Indeed, if arnt1 gene is knocked-out (arnt1$^{-/-}$) an increased GR sensitivity is observed (**Fig 2A'**). To further test whether this had repercussions on steroidogenesis and/or cortisol levels, we analysed them both in a HIF upregulated (vhl$^{-/-}$) and downregulated scenario (arnt1$^{-/-}$).

## HIF signalling acts as negative regulator of steroidogenesis

To investigate the relationship between HIF signalling and steroidogenesis, we initially performed in situ hybridization on larvae obtained from the arnt1$^{+/-}$ mutant line, using both pro-opiomelanocortin (pomca) and Cytochrome P450 family 17 polypeptide 2 (cyp17a2) as probes. Expression of pomca, at the level of the anterior part of the pituitary gland, is a well-established readout of GR function in zebrafish larvae. Pomca is negatively regulated by increased blood cortisol levels via GC-GR signalling, as part of the HPI axis feedback loop [4,51]. Previous

work also suggested that HIF promotes POMC activity in the mouse hypothalamic region [52]. On the other hand, Cyp17a2 is an enzyme involved in steroid hormone biosynthesis at the level of the interrenal gland, which is activated upon ACTH stimulation [53–55].

We found that 5 dpf *arnt1*^-/- larvae, which were characterized by an upregulated GC responsiveness, showed upregulated *cyp17a2* expression (**S3C–S3C' Fig**) coupled to downregulated *pomca* (**Fig 2C**). As expected, *arnt1* siblings showed normally expressed *cyp17a2* (**S3A–S3A' Fig**) and *pomca* (**Fig 2B**), which were observed to be downregulated only as a consequence of BME treatment (**S3B–S3B'** **and S2B'** **Figs**). Therefore, we speculate that in the absence of *arnt1* (HIF suppressed scenario), *pomca* downregulation is most likely to occur as a consequence of GC-GR induced negative feedback loop, triggered by putative high cortisol levels (**Fig 2A and 2G, DMSO, *arnt1* mutant**).

We subsequently examined both *pomca* and *cyp17a2* expression in the opposite -HIF upregulated- scenario, by performing WISH analysis on the *vhl* mutant line. Interestingly, 5 dpf *vhl*^-/- larvae, which were characterized by a downregulated GR activity, displayed downregulated *cyp17a2* expression (**S3G–S3G' Fig**), coupled to downregulated *pomca* expression (**Fig 2E**). On the other hand, *vhl* siblings showed normally expressed *pomca* (**Fig 2D**), which was observed to be downregulated after BME treatment, as expected (**Fig 2D'**). Consequently, we speculate that in the absence of *vhl* (HIF upregulated scenario), *pomca* downregulation is most likely to occur as a consequence of HIF-mediated downregulation of *pomca* expression. (**Fig 2G, DMSO, *vhl* mutant**).

Cumulatively, if this is true, we predicted to observe reduced levels of endogenous cortisol in *vhl*^-/- larvae and normal or even increased levels in *arnt1*^-/- larvae at 5 dpf.

## Steroidogenesis is repressed in *vhl*^-/- and derepressed in *arnt1*^-/-

To confirm this hypothesis, we performed cortisol quantification on the aforementioned *vhl* and *arnt1* mutant lines. Interestingly, cortisol concentration was significantly reduced (P value <0.0028) in *vhl* mutant larvae (92,7 fg/larva), compared to *vhl* siblings (321 fg/larva) (**Fig 2F**). Conversely, cortisol was significantly increased (P value <0.0001) in *arnt1* mutants (487.5 fg/larva), compared to *arnt1* siblings (325 fg/larva) (**Fig 2F'**).

Taken together, these data confirmed our hypothesis and showed for the first time that HIF signalling can act as negative regulator both of GR transcriptional activity and of steroidogenesis. Indeed, if only GR transcriptional activity was blocked by HIF, cortisol levels would be expected to be high in *vhl* mutants. This is because by blocking GR (i.e as occurs in *gr*^-/-), the GC-GR mediated negative feedback cannot occur, making larvae hypercortisolemic [3,5]. Interestingly, since *vhl*^-/- larvae are characterized both by downregulated cortisol levels and GR transcriptional activity, this strongly suggests that HIF signalling can act both at the hypothalamic level (to inhibit *pomca* expression) and intracellularly to block GR transcriptional activity itself.

## Generating *gr* and *gr;vhl* knockout in zebrafish

Conversely, to investigate the role of glucocorticoids on the HIF response, we created a novel glucocorticoid receptor (*gr*, *nr3c1*) mutant line and we crossed it with the *vhl*^hu2117/+^;*phd3:eGFP*^i144/i144^ hypoxia reporter line (this line will be called *gr*^+/-^;*vhl*^+/-^ hereafter). We created this line because the existing *gr*^s357^ allele may still have some activity via non-genomic pathways or tethering, promoting HIF activation upon GC treatment [4,43,51]. Of note, *gr* mutants are hypercortisolemic [3,5]. This is due to the inability of glucocorticoids to bind to a functional receptor (GR). As a result, they fail to provide negative feedback and are not able to shut down GC biosynthesis [3,5]. We generated an 11 bp deletion at the level of *gr* exon 3, which is

predicted to truncate the DNA binding domain, lacks the C-terminal ligand binding domain and is predicted to be a true null (**Fig 3A**). The homozygous *gr/nr3c1* mutants, characterized during the first 5dpf, were morphologically similar to control siblings and adult fish were viable and fertile, as predicted [5].

To confirm loss-of-function, we initially subjected larvae to a visual background adaptation (VBA) test, as it is linked to impaired glucocorticoid biosynthesis and action [4,56]. Larvae derived from *gr*[+/-] incross were VBA tested and sorted according to melanophore size at 5 dpf. PCR-based genotyping on negative VBA-response sorted samples revealed that most larvae were homozygous for the *gr* allele, whereas positive VBA-response samples were always *gr* siblings.

Furthermore, WISH analysis performed on 5 dpf DMSO and BME treated *gr*[+/-] incross derived larvae using *pomca* as probe, showed the presence of upregulated *pomca* expression in DMSO treated gr[-/-] at the level of the anterior part of the pituitary gland (**Fig 3C**), compared to wild-type siblings (**Fig 3B**). Of note, BME treatment was not able to downregulate *pomca* levels in gr[-/-](**Fig 3C'**), as it occurs in BME treated siblings (**Fig 3B'**) via GC-GR mediated negative feedback loop, due to the absence of a functional *gr* allele. Finally, the loss of function was also determined in 5 dpf *gr* mutants by the strong downregulation of *fkbp5* mRNA levels quantified via RTqPCR, both in the presence (fold change = 0.01; P<0.0001) and in the absence of BME treatment (DMSO treated, fold change = 0.01; P<0.0001), compared to DMSO treated wild-types (**Fig 3D**).

## *gr* mutation partially rescues *vhl* phenotype

We next analyzed the effect of *gr* loss of function on *vhl* phenotype. Phenotypic analysis carried out on 5dpf larvae, derived from *gr*[+/-];*vhl*[+/-] incross, revealed that *nr3c1* mutation was able to cause an efficient, but not complete rescue of *vhl* phenotype, in a way which resembled *arnt1* mutation (**Fig 3F' and 3G'**).

In particular, 5dpf *gr*[-/-];*vhl*[-/-] larvae showed a 43% downregulation at the level of the head (P<0.0001), a 66% downregulation in the liver (P<0.0001) and a 51% downregulation in the tail (from the anus to the caudal peduncle) (P = 0.0020), in terms of *phd3:eGFP*-related brightness, compared to *vhl*[-/-] larvae (**Figs 3G' compared to 3E' and S4A**). As expected, 5 dpf double mutant larvae were unable to respond to BME [30 μM] treatment (**Figs 3J–3J' and S4A**), as also confirmed via RTqPCR analysis on HIF (*vegfab* and *egln3*) and GC target genes (*fkbp5*) (**Fig 3H**).

Rescue was also apparent by morphology. Indeed, even if *gr*[-/-];*vhl*[-/-] showed reduced yolk usage, they displayed a reduction in ectopic vessel formation at the level of the dorsal tailfin, no pericardial edema, and developed air-filled swim bladders (**Fig 3G and 3E**). Moreover, whilst *vhl* mutants are inevitably deceased by 10 dpf [47], we were able to raise all selected double mutants beyond 15 dpf, but then (similarly to *arnt1*[-/-];*vhl*[-/-]) they failed to grow and thrive when compared to their siblings. This led us to euthanise them due to health concerns at 21 dpf (**S4B Fig**). Together, these data indicate for the first time, in our *in vivo* animal model, that GR function is essential for HIF signalling in zebrafish larvae, particularly at the level of the head and the liver.

## *gr* loss of function can further reduce HIF signaling in *arnt1;vhl* double mutants

The similarity of *gr* and *arnt1* mutations could mean they work in a single linear "pathway". If true, mutation of both genes should not lead to a further attenuation of the reporter expression. To test this, we bred the *gr* mutant line with the *arnt1;vhl* double mutant line and we

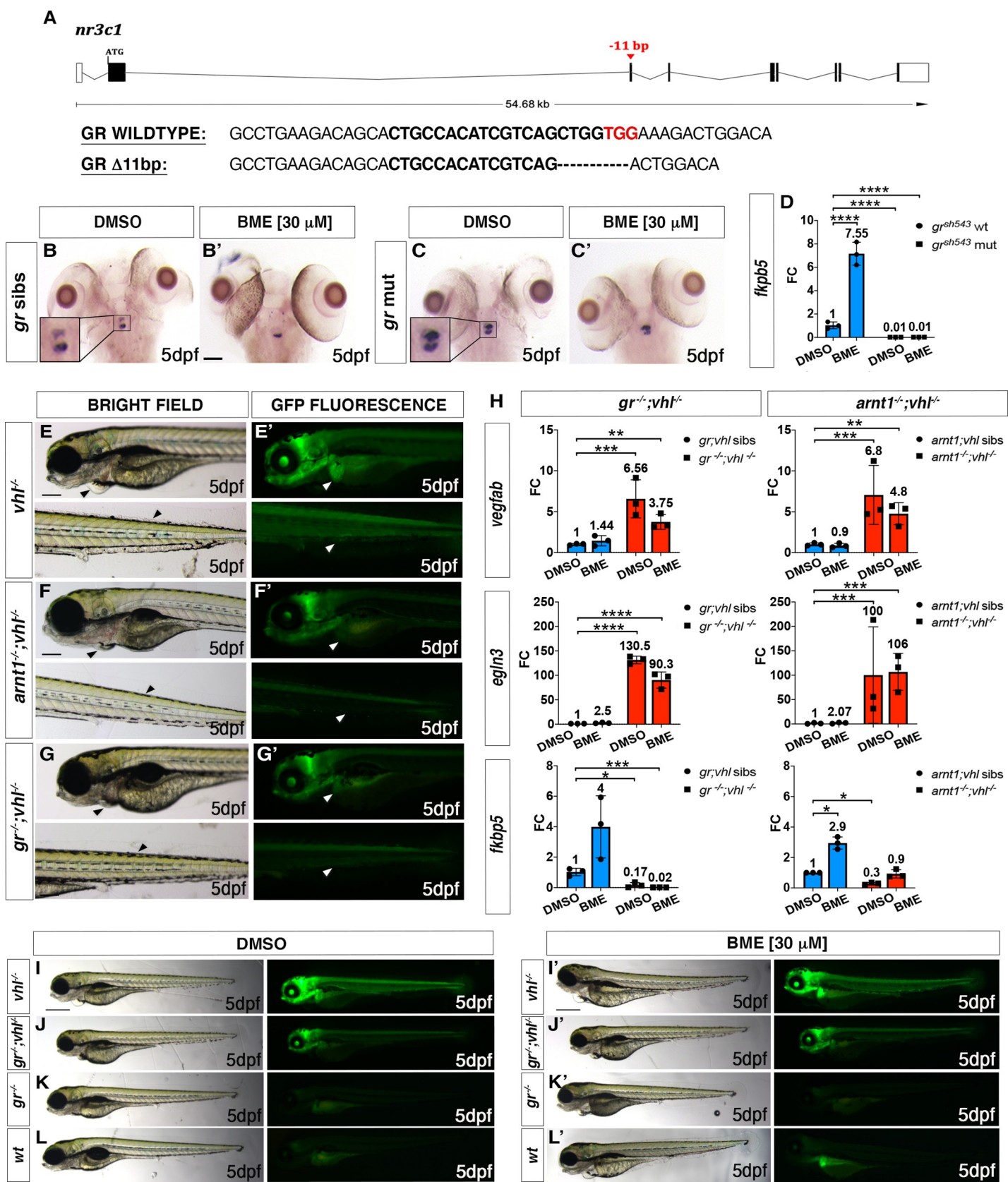

**Fig 3. *gr* mutation partially rescues *vhl* phenotype.** A. Schematic representation of zebrafish *gr (nr3c1)* gene. Exons are shown as boxes, introns as lines. The red arrowhead shows the position of a -11 bp deletion in exon 3 (encoding the DNA binding domain). *gr* wt and mutant sequence. CRISPR target site: bold. PAM sequence: red. B-C'. Representative pictures of WISH performed on DMSO and BME [30 µM] treated *gr* mutant line, at 5 dpf, using *pomca* as probe. Scale bar 100 µm. *gr* siblings DMSO treated (n = 30/30 larvae) showed normal expression; gr siblings (n = 29/30 larvae) showed downregulated pomca expression after BME treatment. Both DMSO treated (n = 30/30) and BME treated (n = 30/30) *gr⁻/⁻* larvae showed upregulated *pomca* expression. D. RTqPCR analysis performed on *gr* wt (n = 10; 3 repeats) and *gr⁻/⁻* (n = 10; 3 repeats) larvae at 5 dpf, using *fkbp5* as probe. Statistical analysis was performed on ΔΔCt values, whereas data are shown as fold change values. Ordinary Two-way ANOVA followed by Dunnett's multiple comparison test (****P < 0.0001). E-G. Magnified picture of representative *gr⁻/⁻; vhl⁻/⁻* larvae compared to *arnt1⁻/⁻;vhl⁻/⁻* and *vhl⁻/⁻* larvae. Both double mutants are characterized by the absence of pericardial oedema, no ectopic extra vasculature at the level of the tail, no bright liver and a reduced brightness in the rest of the body (white and black arrowheads), compared to *vhl⁻/⁻* larvae. Fluorescence, exposure = 2 seconds. Scale bar 200 µm. H. RTqPCR analysis performed both on HIF and GC target genes expression carried out on *gr⁻/⁻; vhl⁻/⁻* and sibling at 5 dpf, (n = 10 larvae, per group, in triplicate) compared to *arnt1⁻/⁻;vhl⁻/⁻* larvae and siblings, at 5dpd (n = 10 larvae, per group, in triplicate). Both *vegfab* and *egln3* are HIF target genes, whereas *fkbp5* is a GC target gene. Statistical analysis was performed on ΔΔCt values, whereas data are shown as fold change values, Ordinary Two-way ANOVA followed by Dunnett's multiple comparison test. I-L. Representative picture of phenotypic analysis performed on DMSO and BME [30 µM] treated *gr⁺/⁻; vhl⁺/⁻(phd3:eGFP)* incross-derived 5 dpf larvae (n = 600). All the genotype combinations observed are represented in the figure. Among the 450 GFP⁺ larvae analysed, 28 showed a partially rescued *vhl* phenotype which resembled the arnt1's one. Three experimental repeats. In all panels: *P < 0.05; **P < 0.01; ***P <0.001; ****P < 0.0001. Fluorescence, exposure = 2 seconds. Scale bar 500 µm.

crossed *gr⁺/⁻;arnt1⁺/⁻;vhl⁺/⁻* triple carriers. Phenotypic analysis carried out on 5 dpf *phd3:eGFP* positive larvae (n = 488) showed a small class of larvae with an even more rescued phenotype and a stronger downregulation of *phd3:eGFP* related brightness compared both to *arnt1⁻/⁻; vhl⁻/⁻* (**Fig 4B–4B'**) and *gr⁻/⁻;vhl⁻/⁻* double mutants (**Fig 4C–4C'**). Of note, 7 putative very weak GFP⁺ larvae were selected and genotypic analysis confirmed that 5 out of 7 were indeed *gr⁻/⁻; arnt1⁻/⁻;vhl⁻/⁻*. In particular, these triple mutants showed a 54% downregulation at the level of the head, a 71% downregulation in the liver and a 72% downregulation in the tail region, in terms of *phd3:eGFP*-related brightness compared to *vhl⁻/⁻* (**Figs 4D–4D' and S5**). Thus, these data suggest that glucocorticoids are likely to act on both Arnt1 and Arnt2 mediated HIF signalling pathway.

## The BME-induced HIF response is Arnt1 dependent

To further examine the effect of glucocorticoids on HIF signalling, we performed BME [30 µM] treatment on all the available mutant lines. Of note, unlike cortisol, betamethasone has a very high affinity for Gr, but an insignificant affinity for Mr [57,58]. As expected, 5 dpf wild-types larvae showed a mild upregulation of *phd3:eGFP*-related brightness at the hepatic level, compared to untreated controls (**Figs 1G' and 3L'**). BME treatment was also able to further increase *phd3:eGFP*-related brightness at the level of the head and the liver of 5 dpf *vhl⁻/⁻*, as also confirmed by WISH, using both lactate dehydrogenase A (*ldha*) (**Fig 5B–5B', black arrowheads**) and prolyl hydroxylase 3 (*phd3*) as probes (**Fig 5D–5D', black arrowheads**). As predicted, both *gr⁻/⁻* and *gr⁻/⁻;vhl⁻/⁻* mutants were unaffected due to the absence of functional Gr (**Fig 3K–3K' and 3J–3J'**). Interestingly, in both the *arnt1⁻/⁻*(**Fig 1F**) and also *arnt1⁻/⁻;vhl⁻/⁻* (**Fig 1E**) the *phd3:eGFP*-related brightness did not change after BME treatment (**Fig 1E' and 1F'**). This was also confirmed via, RTqPCR analysis carried out on HIF target gene *egln3/phd3* in the *arnt1* mutant (**S1C Fig**; see also *arnt1⁻/⁻;vhl⁻/⁻* **Fig 3H**)

Taken together these data suggest that in *vhl⁻/⁻* larvae, BME treatment can upregulate HIF signalling by "bypassing" HIF-mediated *pomca* negative regulation. By directly binding to Gr, it can compensate for the repressed cortisol levels in *vhl* mutants. (**Fig 2F**). On the other hand, both in *arnt1⁻/⁻* and also *arnt1⁻/⁻;vhl⁻/⁻* larvae, even if BME can act downstream of *pomca*, it can only upregulate GR responsiveness, but cannot upregulate HIF signalling due to *arnt1* loss of function. Cumulatively, we speculate that even if Arnt2 can interact with the HIF-α isoforms to maintain a moderately upregulated HIF levels (*arnt1⁻/⁻;vhl⁻/⁻*), the BME-mediated HIF upregulation is Arnt1 dependent.

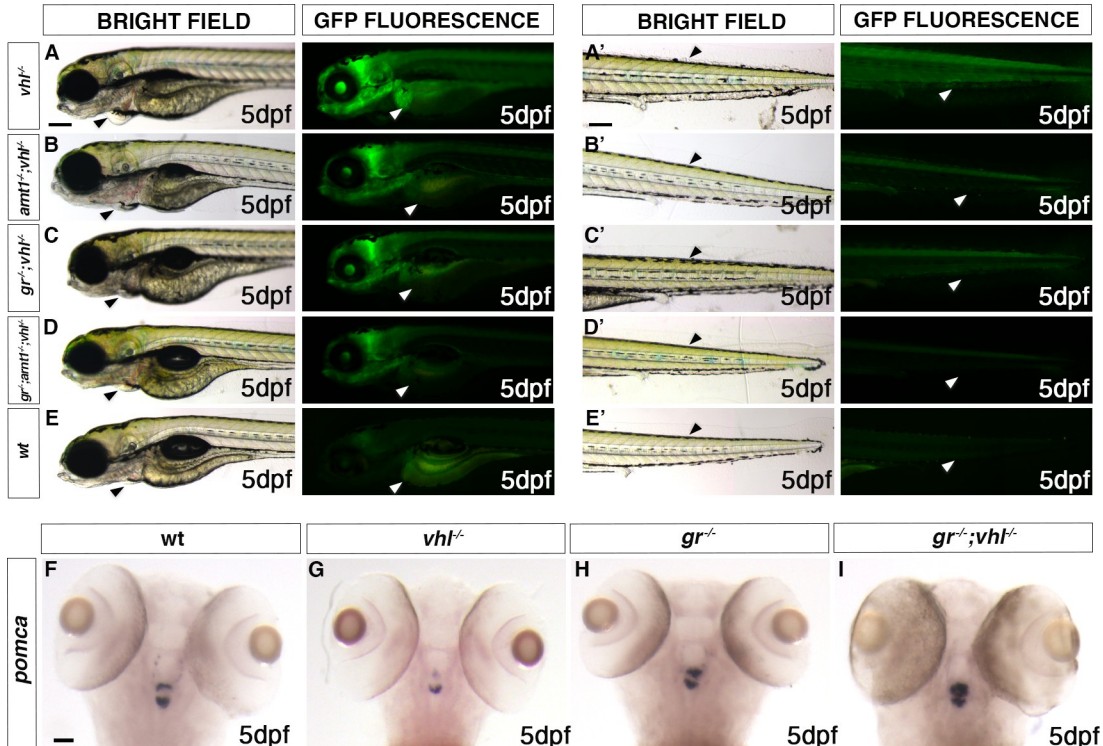

**Fig 4. *gr* loss of function effect is stronger when HIF-signalling is moderately upregulated.** A-E. Representative picture of the main differences between *vhl*⁻/⁻, *arnt1*⁻/⁻;*vhl*⁻/⁻, *gr*⁻/⁻;*vhl*⁻/⁻ and triple *gr*⁻/⁻;*arnt1*⁻/⁻;*vhl*⁻/⁻ larvae at 5 dpf. Among the 488 *phd3*:*eGFP* expressing larvae analysed, 7 larvae were characterized by the absence of pericardial oedema (black arrowheads, left), no ectopic extra vasculature at the level of the tail (black arrowheads, right), no visible *phd3*:*eGFP* HIF reporter in the liver (white arrowheads, left) and even more reduced levels of this marker in the head and in the rest of the body (white arrowheads, right). Genotypic analysis allowed to confirm the presence of a genotype-phenotype correlation in 5 out 7 samples and to prove that they were triple mutants. Fluorescence, exposure = 2 seconds. Scale bar 200 μm. F-I. Representative pictures of WISH performed on *gr*⁺/⁻; *vhl*⁺/⁻ incross derived larvae, at 5 dpf, using *pomca* as probe. Of note, *gr*⁻/⁻;*vhl*⁻/⁻ showed upregulated *pomca* expression (20/20 larvae), as observed in *gr*⁻/⁻ (20/20 larvae); *vhl* mutants showed downregulated *pomca* expression (20/20 larvae), whereas wildtypes showed normal pomca expression (19/20). Chi-square test (****P < 0.0001). Scale bar 50 μm.

### *gr* mutation overrides HIF-mediated *pomca* suppression in a *vhl* deficient background

To examine the effect of *gr* loss of function on steroidogenesis in *gr*⁻/⁻;*vhl*⁻/⁻, we performed WISH analysis on 5 dpf *gr*⁺/⁻;*vhl*⁺/⁻ incross derived larvae, using *pomca* as probe. As expected, *vhl*⁻/⁻ showed downregulated *pomca* expression (**Fig 4G**), whereas *gr* ⁻/⁻ displayed upregulated *pomca* (**Fig 4H**), compared to wildtypes (**Fig 4F**). Notably, since a strong upregulation of *pro-opiomelanocortin a* was observed in the double mutants (**Fig 4I**), this suggests that *gr* mutation overrides HIF-mediated *pomca* inhibition. PCR-analysis performed post-WISH confirmed this genotype-phenotype correlation.

These data, in accordance with our hypothesis, suggest that in *gr*⁻/⁻;*vhl*⁻/⁻ mutants the up-regulation of *pomca*, triggered by the absence of functional Gr (and of the GC-Gr mediated negative feedback), cannot be inhibited with the same efficiency by HIF activity at the hypothalamic level. In *gr*⁻/⁻;*vhl*⁻/⁻ mutants, we speculate that the upregulated endogenous cortisol interacts with Mr to stimulate the HIF pathway, resulting in a mildly upregulated *phd3*:*EGFP* expression, in-between the levels seen in *vhl* mutants and wild-type larvae (**Fig 3J and 3I***)*. To test this assumption, we set up to block the *mr gene* in a *gr*⁻/⁻;*vhl*⁻/⁻ background, in order to check the importance of Mr in the HIF signaling pathway.

## Both Gr and Mr are directly required in the HIF signalling pathway

Cortisol has high affinity both for Gr and Mr and they have been recently shown to be differentially involved in the regulation of stress axis activation and function in zebrafish [3]. Therefore, we analysed the role of Mr on the HIF signaling pathway. To achieve this, we knocked-out *mr* in *gr*$^{+/-}$*;vhl*$^{+/-}$*;phd3:eGFP* incross-derived embryos, using CRISPant technology [48,59]. Interestingly, phenotypic analysis performed on 5 dpf injected and uninjected larvae revealed that *mr* CRISPR injected *vhl* mutants were characterized by a significant downregulation of *phd3:eGFP*-related brightness at the level of the head (equals to 49%, P<0.0001), in the liver (equals to 56%, P<0.0001) and in the rest of the body (equals to 47%, P<0.0001), compared to *vhl*$^{-/-}$ mutant uninjected larvae (**Fig 6D compared to 6A**). Moreover, when both *gr* and *mr* were knocked-out, the downregulation was even stronger at the level of the head (equals to 62%, P<0.0001), in the liver (equals to 77%, P<0.0001) and in the rest of the body (equals to 63%, P<0.0001) compared to *vhl*$^{-/-}$ mutant uninjected larvae (**Fig 6E compared to 6A**). Of note, *mr* injection in *vhl*$^{-/-}$ larvae was more efficient in downregulation of *phd3:eGFP* expression compared to uninjected *gr*$^{-/-}$*;vhl*$^{-/-}$ larvae at the level of the head (equals to 31%, P = 0.0087) (**Fig 6D compared to 6B**).

To test the reliability of CRISPant method, we chose to knock-out a gene (which was not involved in the HIF pathway) into *vhl*$^{+/-}$ incross derived embryos, to test whether it was able to affect HIF signalling. *Laminin, beta 1b* (*lamb1b*), which codes for an extracellular matrix glycoprotein, was injected as CRISPR-injection control in *vhl*$^{+/-}$incross derived embryos at 1 cell stage. Genotypic analysis carried out on these larvae confirmed that these guides were effective. Finally, quantification of *phd3:eGFP*-related brightness performed on 5 dpf injected and uninjected *vhl*$^{-/-}$ larvae, showed no significant differences between the two groups (**S6A and S6C Fig**). Overall, these data corroborated the efficiency of the CRISPant method and, at the same time, confirmed that both glucocorticoid and mineralocorticoid receptor play a pivotal role in the HIF signalling *in vivo*.

## Discussion

Both HIF and glucocorticoid mediated transcriptional responses play a pivotal role in tissue homeostasis, glucose metabolism and in the regulation of cellular responses to various forms of stress and inflammation [60–62]. Previous *in vitro* studies highlighted the potential for crosstalk between HIF and glucocorticoid pathways, however there are still conflicting data on how this interaction occurs *in vivo* and there is no information on Mr contribution to HIF signalling. In this regard, we have presented a novel *in vivo* study using zebrafish larvae, focusing on the crosstalk between these two pathways. In contrast to *in vitro* cell culture studies, a whole animal study allows us to consider the interactions that occur between various tissues and provide novel insights. To this end, we generated *arnt1* and *gr* null mutants to downregulate HIF and GR signalling respectively, as a basis for a genetic analysis of this crosstalk.

As a prelude to this, we had to establish the relative importance of *arnt1* and *arnt2* in the overall HIF response. To achieve this, a discriminative test was devised to place them in a *vhl* mutant background, where HIF signaling is strongly upregulated [25,42]. Phenotypic analysis performed on 5 dpf *arnt1*$^{-/-}$*;vhl*$^{-/-}$ larvae showed reduced *phd3:eGFP* related brightness, normal yolk usage, properly developed and air-filled swim bladder as well as by the absence of pericardial oedema and excessive caudal vasculature. However, beyond 5 days, these double mutants exhibited only partial recovery from the *vhl* phenotype. Indeed, they developed well till 15 dpf, but subsequently failed to grow and thrive when compared to their siblings. In addition, *arnt1* homozygous mutants were found to be viable and fertile, in contrast to both homozygous *vhl* and *arnt2* mutants, which are embryonic lethal by 8–10 dpf [14,47].

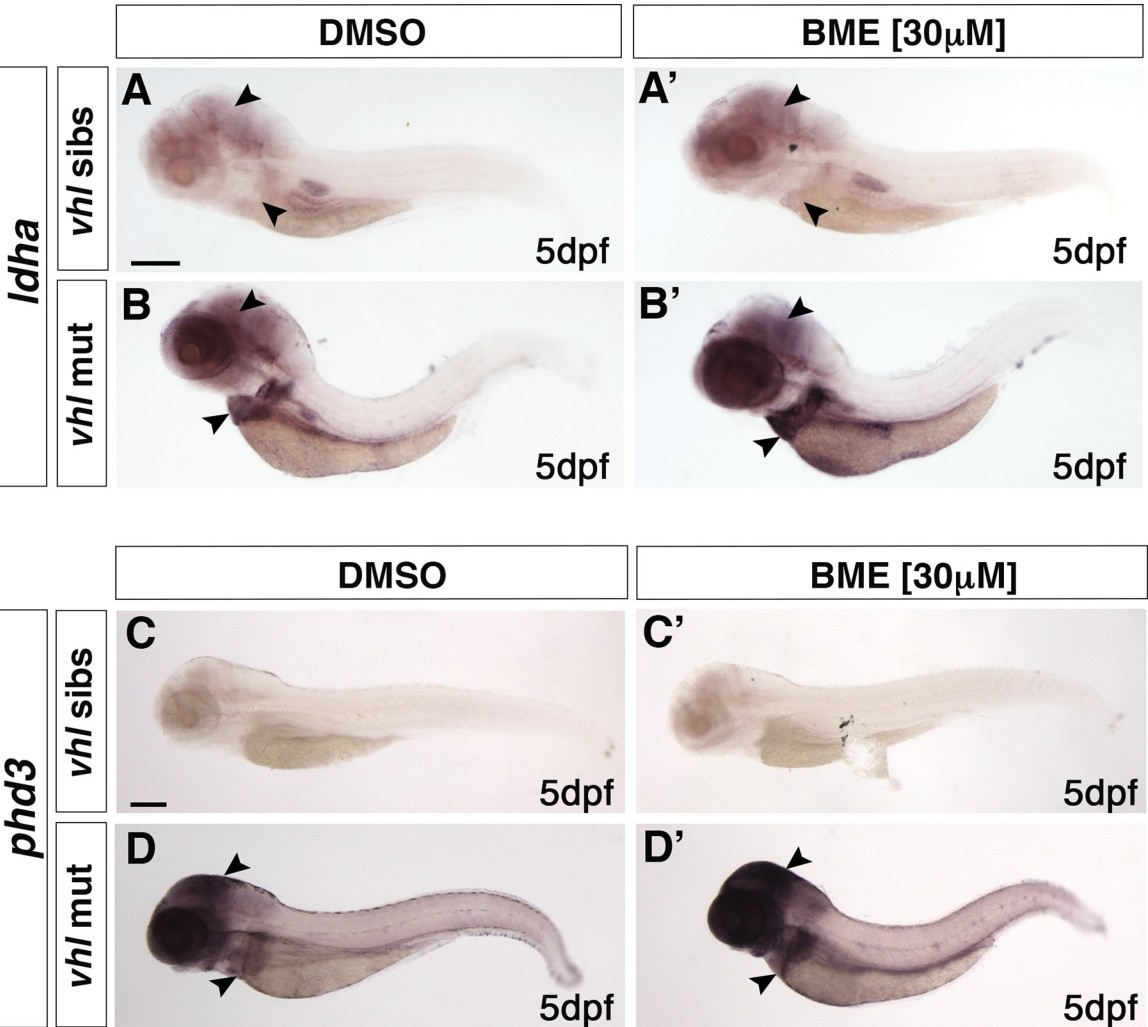

**Fig 5. BME treatment is able to upregulate HIF signalling in *vhl*-/-.** A-B'. Representative pictures of WISH performed on DMSO (A-B) and BME [30 μM] (A'-B') treated *vhl*+/- incross derived larvae, at 5 dpf, using *ldha* as probe. DMSO treated *vhl* siblings showed basal *ldha* expression (34/35 larvae), which showed to be upregulated after BME treatment (33/35 larvae). On the other hand, DMSO treated *vhl*-/- showed upregulated *ldha* expression (32/35 larvae), which was further upregulated after BME treatment (34/35 larvae). (black arrowhead: head and liver) Chi-square test (****P < 0.0001). Scale bar 200 μm. C-D'. Representative pictures of WISH performed on DMSO (C-D) and BME [30 μM] (C'-D') treated *vhl*+/- incross derived larvae, at 5 dpf, using *phd3 (egln3)* as probe. As expected, *vhl* siblings DMSO treated (n = 30/30 larvae) showed basal *phd3* expression, which was mildly increased after BME treatment (n = 27/30 larvae). *Vhl*-/- DMSO treated (n = 28/30 larvae) showed upregulated *phd3* expression, which was further increased after BME treatment (n = 26/30 larvae). (black arrowhead: head and liver) Chi-square test (****P < 0.0001). Scale bar 200 μm.

Even though Arnt1 is not fundamental for survival, we found that it is required in the liver and in organs outside the central nervous system for HIF−α function. Conversely, using CRIS-Pant technology [48,59], we established that Arnt2 is mainly required in the developing central nervous system (CNS), as also reported by Hill et al. in 2009 [14]. However, the similarities observed in terms of *phd3:eGFP*-induced brightness in both *arnt1*-/-;*vhl*-/- and *arnt2* CRISPR injected *vhl* mutants, suggest there is no strong functional separation. Therefore, both Arnt2 and Arnt1 have partially overlapping functions *in vivo* and both contribute to the HIF response.

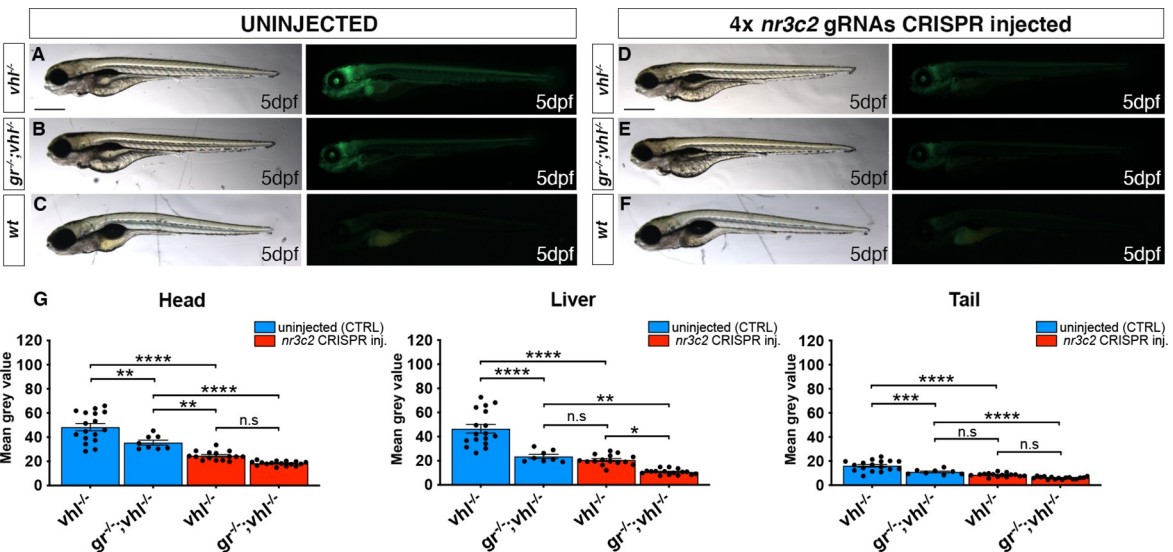

**Fig 6. Both Gr and Mr are directly required in the HIF signalling pathway.** A-F. Representative pictures of 5 dpf CRISPANT mutants created by redundantly targeting *nr3c2* (*mr*) gene via co-injection of 4x gRNAs in *gr*^+/-;*vhl*^+/-(*phd3:eGFP*) x *gr*^-/-; *vhl*^+/-(*phd3:eGFP*) derived embryos (n = 344). Uninjected embryos were used as control (n = 170). Fluorescence, exposure = 991,4 ms. Scale bar 500 μm. G. Statistical analysis performed on mean grey value quantification (at the level of the head, liver and tail), after phenotypic analysis, on 5 dpf *mr* 4x gRNAs injected and uninjected larvae. *vhl*^-/- uninjected n = 17 larvae: head 48.28 ± 2.99 (mean ± s.e.m); liver 46.47 ± 3.55 (mean ± s.e.m); tail 16.15 ± 1.06 (mean ± s.e.m). *gr*^-/-;*vhl*^-/- uninjected n = 8 larvae: head 35.48 ± 2.03 (mean ± s.e.m); liver 23.56 ± 1.72 (mean ± s.e.m); tail 10.98 ± 0.75 (mean ± s.e.m). *vhl*^-/- injected n = 15 larvae: head 24.62 ± 0.97 (mean ± s.e.m); liver 20.67 ± 1.1 (mean ± s.e.m); tail 8.57 ± 0.39 (mean ± s.e.m). *gr*^-/-;*vhl*^-/- injected n = 16 larvae: head 18.33 ± 0.46 (mean ± s.e.m); liver 10.71 ± 0.56 (mean ± s.e.m); tail 6.07 ± 0.26 (mean ± s.e.m); ordinary One-way ANOVA followed by Sidak's multiple comparison test.

## The effect of HIF signalling on the glucocorticoid pathway

We next investigated the effect of HIF signalling and glucocorticoid responsiveness, by performing RTqPCR analysis on 5 dpf larvae. Collectively, we show that strong activation of HIF signalling (in *vhl*^-/-) is able to blunt glucocorticoid receptor transcriptional regulation as judged by *fkbp5* expression, whereas *arnt1* loss of function derepressed it. As our experiments are done at normal atmospheric oxygen levels, we conclude from the latter result that normoxic HIF activity nevertheless suffices to attenuate GR transcriptional regulation.

We checked whether HIF signalling affects steroidogenesis. To this end, we quantified the expression of steroidogenesis-related genes (*pomca* and *cyp17a2*) both in *vhl*^-/- and in *arnt1*^-/- larvae, via whole-mount *in situ* hybridization. Surprisingly, both lines showed downregulation of *pomca* expression. However, *arnt1*^-/- larvae showed upregulated *cyp17a2* expression, whereas *vhl*^-/- larvae, were characterized by downregulated *cyp17a2*.

Considering our results with GR-target *fkbp5* in these mutants, we assume that in an *arnt1* knock-out scenario, *pomca* downregulation occurs as a consequence of the GC/GR-mediated negative feedback loop aimed to control cortisol biosynthesis. This is also consistent with a significant upregulated basal cortisol levels quantified in these mutants. Vice versa, when HIF signalling is upregulated (in *vhl* mutants) we speculate that *pomca* and cyp17a2 downregulation may occur via HIF-mediated activity, leading to the observed low cortisol levels coupled to suppressed GR activity.

Indeed, glucocorticoids regulate a plethora of physiological processes, act on nearly every tissue and organ in the body to maintain homeostasis and are characterized by a potent anti-inflammatory and immunosuppressive actions. For these reasons, their secretion must be finely controlled by the HPA/I axis [63].

As previous work in our laboratory showed that glucocorticoids also act as HIF activators [25,43], we infer that HIF can in turn control GC levels by acting on *pomca*. This would enable HIF signalling not only to control its own levels, but also to assure homeostasis. Finally, since HIF signalling is a master regulator of cellular pro-inflammatory responses to hypoxia [64–66], which would counteract the anti-inflammatory glucocorticoid activity, we speculate that the simultaneous expression of both upregulated HIF and GC pathway would be detrimental to homeostasis.

Our data would also be in accordance with a previous study showing that hypoxia exposure resulted in downregulation of steroidogenic genes (*StAR*, *cyp11c1*, *hmgcr*, *hsd17b2*, *cyp19a*, *cyp19b*) in 72 hpf larvae, whereas zHIF-α loss of function triggered the upregulation specifically of *StAR*, *cyp11b2* and *cyp17a1* [67].

Cumulatively, if this is true, we predicted to observe reduced levels of endogenous glucocorticoids in *vhl*$^{-/-}$ and normal or even increased levels in *arnt1*$^{-/-}$. Importantly, the fact that cortisol levels were lowered in *vhl* mutants and were upregulated in *arnt1* mutants is consistent with our hypothesis.

As a consequence of the above considerations, the HIF-mediated *pomca* negative regulation seems to be a logic homeostatic interaction: Increased HIF reduces GR activity, which in turn should lead to less HIF signalling.

## The effect of glucocorticoids on the HIF signaling pathway

To further investigate the role of glucocorticoids on the HIF signalling, we initially analyzed the effect of *gr* loss of function on *vhl* phenotype. Surprisingly, we observed that *gr* mutation was able to cause an efficient, but not complete rescue of the *vhl* phenotype. Notably, *gr*$^{-/-}$;*vhl*$^{-/-}$ survived much longer than *vhl*$^{-/-}$ (> = 21 dpf compared to max. 10 dpf), but then similar to *arnt1*$^{-/-}$;*vhl*$^{-/-}$, they failed to grow and thrive when compared to siblings. Our previous work [43] established that activation of GR signalling negatively regulates VHL protein in human liver cells. Our current genetic analysis shows that in zebrafish larvae, there must be an additional point of interaction between these two pathways, as we observed further activation of our HIF reporter after GC treatment even in the absence of VHL. Cumulatively, we showed for the first time in an *in vivo* animal model that Gr is fundamental to allow high HIF signalling levels.

We next analysed the effect of betamethasone treatment in *arnt1*$^{-/-}$. Although BME activated the GR target *fkbp5*, as expected, it failed to activate HIF signaling [43]. This was unexpected and would be best explained by assuming that a Gr-BME complex would preferentially interact with a HIFα/ARNT1 complex but not a HIFα/ARNT2 complex. Whether this holds up in mammalian cells would be interesting to address.

## Evaluation of mineralocorticoid receptor contribution to HIF signalling

Recent work published by Faught and Vijayan, 2018 showed that both Gr and Mr are involved in the regulation of zebrafish stress axis activation and function [3]. Nothing is known about mineralocorticoid receptor contribution to HIF signalling. Therefore, we tested the effect of *mr* knock-out in *gr*$^{+/-}$;*vhl*$^{+/-}$;*phd3:eGFP* incrossed derived embryos. Interestingly, in *mr* injected- *vhl*$^{-/-}$ we observed a significant reduction of *phd3:eGFP*-related brightness, compared to uninjected *vhl*$^{-/-}$ larvae. Moreover, a further reduction of *phd3:eGFP* expression was found at the level of the head in *mr* injected- *vhl*$^{-/-}$ compared to *gr*$^{-/-}$;*vhl*$^{-/-}$ larvae. Finally, the additional removal of *mr* in a *gr*$^{-/-}$;*vhl*$^{-/-}$ background reduced the hypoxia reporter expression even further.

Therefore, we were able to show that both the glucocorticoid receptor and mineralocorticoid receptor play a pivotal role in promoting HIF signaling in zebrafish. In contrast to mammals, teleosts lack aldosterone and cortisol is the primary glucocorticoid hormone that can interact both with Gr and Mr to assure a correct HPI axis activity [68,69]. Of note, Mr was shown to not have a role in rapid non-genomic behaviors that required HPI axis signaling in zebrafish [70]. However, our hypothesis is consistent with Faught and Vijayan, 2018 elegant work, showing that both Gr and Mr signalling are involved in the GC negative feedback regulation. Importantly, this outcome may have a wider significance in health and disease. This is because so far, HIF signalling, which plays a key role in tumour growth, is proven difficult to downregulate. In this regard, our study suggests that modulation of Gr and Mr might be a potential avenue. In conclusion, although Mr contribution to HIF response in other organisms remains unclear, our work suggests that research into its function is warranted.

## Conclusion

Our present study stresses the importance of the glucocorticoid pathway in driving HIF signalling. In addition, we uncovered a negative regulatory role played by HIF in regulating both GR responsiveness and steroidogenesis as demonstrated via RTqPCR and steroid hormone quantification. We also identified a mineralocorticoid receptor contribution to HIF-GC crosstalk. Finally, we presented novel $gr^{+/-};vhl^{+/-}$, $arnt1^{+/-};vhl^{+/-}$ and $arnt1^{+/-};gr^{+/-};vhl^{+/-}$ zebrafish mutant lines which helped to better understand how the interplay between HIF and glucocorticoids occurs *in vivo*. For these reasons, we believe that this work could pave the way for further *in vivo* analysis to precisely identify the extensive crosstalk behind these two major signalling pathways.

## Materials and methods

### Ethics statement

Zebrafish (Danio rerio) lines were raised and maintained under standard conditions (14 hours of light and 10 hours of dark cycle, at 28˚C) in the Aquaria facility of the University of Sheffield. Zebrafish embryos used for experiments were reared in E3 medium (5 mM NaCl, 0.17 mM KCl, 0.33 mM MgCl2, 0.33 mM CaCl2, pH 7.2) with or without methylene blue (Sigma-Aldrich) and staged according to standard methods [71] for up to 5,2 days post fertilisation (dpf) in accordance with UK Home Office legislation. Our studies conform with the UK Home Office guidelines (ASPA), Licence No. PC9C3D4CB and PB2866ED0. Ethics approval was obtained from the University of Sheffield Ethics committee AWERB.

### Zebrafish strains

The following zebrafish lines were used: wild-type (wt) strain AB (ZDB-GENO-960809-7), $vhl^{hu2117/+};phd3$:eGFP$^{i144/i144}$ (ZDB-GENO-090611-18), $hif1\beta^{sh544/+}$; $hif1\beta^{sh544/+};vhl^{hu2117/+}$, $gr^{sh543/+}$, $gr^{sh543/+};vhl^{hu2117/+}$, $gr^{sh543/+};hif1\beta^{sh544/+}$; and $gr^{sh543/+};hif1\beta^{sh544/+};vhl^{hu2117/+}$ lines were generally maintained in a $phd3$:eGFP$^{i144/+}$ background. The following 4x gRNAs CRISPR-injected G0 null mutant lines were created according to Wu et al, 2018 [48] protocol and raised up to to 5,2 dpf: $mr;gr^{sh543/+};vhl^{hu2117/+};phd3$:eGFP$^{i144/+}$, $hif1\beta2;1hif1\beta^{sh544/+}$; $vhl^{hu2117/+};phd3$:eGFP$^{i144/+}$ and $lamb1b;vhl^{hu2117/+};phd3$:eGFP$^{i144/+}$.

### Generation of *gr* (*nr3c1*) and *hif1β* (*arnt1*) null zebrafish lines

Both *nr3c1* mutant line ($gr^{sh543/+}$) and *arnt1* mutant line ($hif1\beta^{sh544/+}$) were generated using the CRISPR/Cas9-based mutagenesis method. A gene-specific guide RNA (sgRNA) sequence was

identified using the CHOPCHOP website [72,73]. To design both *gr* and *arnt1* sgRNA, an 18 nucleotides sequence upstream to a selected PAM site (*gr*<sup>sh543</sup>: CCAGCTGACGATGTGG CAG; *hif1β*<sup>sh544</sup>: TCGGTGCTGGTGTTTCCAG) was inserted into a scaffold sequence [74], containing a promoter for the T7 Polymerase. The sgRNA was amplified via PCR, purified from agarose gel and *in vitro* transcribed using MEGAshortscript T7 kit (Ambion). 1 nl of CRISPR mixture containing 2,4 $\mu$g/$\mu$l of gRNA and 0.5 $\mu$l Cas9 protein (NEB) was injected in one-cell stage embryos and raised for 24 hours. Wild-type (wt), strain AB embryos were used to generate the *gr* mutant line, whereas *vhl*<sup>hu2117/+</sup>;*phd3:eGFP*<sup>i144/+</sup> incross-derived embryos were used to create the *hif1β* mutant line. Efficiency was verified via whole-embryo PCR-based genotyping, by a diagnostic restriction digest. Injected embryos were raised to adulthood. Embryos collected from transmitting G0 founders crossed with WT(AB) fish were raised and genotyped to confirm germline transmission of the mutation (F1 generation). Heterozygous mutants, carrying the same mutation, were selected and crossed to obtain homozygous mutant embryos (F2 generation).

## Generation of CRISPR/Cas9-mediated mutants (CRISPANTs)

To generate G0 knockout embryos we used the method developed by Burger et al 2016 [59] and improved by Wu et al., 2018 [48]. In short, a pool of four guide-RNAs (25$\mu$M each, Sigma Aldrich) were co-injected with 0.5 $\mu$l Cas9 protein (NEB, M0386, 20$\mu$M), diluted 1:10) and 1 $\mu$l tracrRNA (100$\mu$M) in one-cell stage embryos. This method was used to create G0 CRISPANTs for the following genes of interest: *mineralocorticoid receptor* (*mr*, *nr3c2*), *aryl hydrocarbon receptor nuclear translocator 2* (*arnt2*, *hif1β2*) and *laminin, beta 1b* (*lamb1b*). The latter was used as CRISPR-injection control. The gRNA target sequences used in this study are as follows: *arnt2*: gRNA1-ACGGCGCCTACAAACCCTCC (exon 5), gRNA2-GGCCGATGGC TTCTTGTTCG (exon6), gRNA3-TTCACGCCACAATTCGGATG (exon11), gRNA4-GT CGCAGGTGCGTAAAAACA (exon 14); *nr3c2*: gRNA1-GCATTGTGGGGTCACCTCCA (exon 2), gRNA2-AAGGGGATTAAACAGGAAAC (exon 2), gRNA3-CAACCAGCTCGC CGGAAAAC (exon 5), gRNA4-ATATCTGACGCCGTCCGTCT (exon 5); *lamb1b* gRNA1- TTGTTAATAGCATAGTACATTGG (sequence upstream 5'UTR), gRNA2-GGAGAACAA GCAAAACGATGAGG (ATG), gRNA3- GCGTGGTGCAGGGTTTGTAG (5'UTR), gRNA4- TCACAATGACATGTGTGCG (exon 2). The success of the injection was determined via phenotypic analysis, followed by quantification of *phd3:eGFP* related brightness and whole-embryo PCR-based genotyping performed on a fraction of injected embryos at 5 dpf.

## Whole-mount *in situ* hybridisation

Whole-mount *in situ* hybridization (WISH) was performed according to standard protocols [75]. The following antisense RNA probes were used: *proopiomelanocortin a* (*pomca*) created as previously described [76]; *Cytochrome P450 family 17 polypeptide 2* (*cyp17a2*), created as previously described [55], both *prolyl hydroxylase 3* (*phd3; BC066699*), and *lactate dehydrogenase A* (*ldha1; BC067188)* probes, generated as previously described [25,47].

## Embryos harvesting, drug treatment and fixation for WISH

Embryos intended for whole-mount *in situ* hybridisation were treated with 16,8 $\mu$l of 1-phenyl 2-thiourea (PTU, stock concentration 75mg/ml) diluted into 35 ml E3 medium to inhibit melanogenesis, according to Karlsson et al., 2001 [77]. GR agonist treatment was performed on batches of 15 embryos each, at 4 dpf, treated in 6-well plates, with 30 $\mu$M Betamethasone 17,21-dipropanoate (BME) and with 1% DMSO (Sigma-Aldrich), as control, for 24 hours [4]. Inside the 6-well plates, embryos were incubated in 3 ml total volume of E3 medium, without

methylene blue. Afterwards, up to 30 embryos at 5 dpf were collected in 1,5 ml Eppendorf tubes and anaesthetized using Tricaine Solution (MS-222, Sigma Aldrich) prior to fixation in 1 ml 4% PFA solution overnight, at 4˚C. Embryos were then washed twice for 10 minutes in PBST and post-fixed in 1 ml 100% MeOH. Finally, samples were stored at -20˚C.

### $gr^{sh543}$ mutants sorting by visual background adaptation (VBA)

Visual background adaptation (VBA) is a glucocorticoid receptor-dependent neuroendocrine response which causes zebrafish melanocytes to shrink when exposed to bright illumination [78,79]. To identify $gr^{sh543}$ mutants from siblings and to confirm the absence of a functional VBA response, 5dpf larvae were exposed to 30 minutes darkness and then transferred onto a white background under bright, whole-field illumination, using a 30W fluorescent lamp mounted 50 cm above the dish [80,81].

### Cortisol extraction and quantification

Three biological replicates of 150 larvae at 5 dpf each of $hif1\beta^{sh544}$ mutants, $hif1\beta^{sh544}$ siblings, $vhl^{hu2117}$ mutants and $vhl^{hu2117}$ siblings, respectively, were used for steroid hormone extraction and quantification. $vhl^{-/-}$ larvae were sorted among siblings at 4 dpf according to both their phenotype and $phd3$:$eGFP$-related brightness. Because of the lack of visible phenotype, $arnt1^{-/-}$ larvae where derived from $arnt1^{-/-}$ fish incrossed, whereas siblings were from $arnt1^{+/-}$ fish crossed with $arnt1^{+/+}$ ones. Cortisol quantification was carried out according to the protocol published by Eachus $et$ $al.$, 2017 [55], based on the use of an Acquity UPLC System (Waters, Milford, CT) coupled to a Xevo TQ-S tandem mass spectrometer (Waters).

### RNA isolation, cDNA synthesis and qPCR analysis

Transcript abundance of target genes was measured by quantitative real-time PCR (RTqPCR). Three biological replicates of 10 larvae at 4 dpf each, were treated for 24 hours with 30 μM Betamethasone 17,21-dipropanoate and with 1% DMSO, used as control, prior to RNA isolation. Total RNA was extracted from pools of 10 larvae at 5dpf with TRIzol reagent (Invitrogen by Thermo Fisher Scientific, 15596026). RNA extracted was quantified using a Nanodrop ND-1000 spectrophotometer. cDNA was then synthesized from 1μg RNA template through reverse transcription using Protoscript II First Strand cDNA Synthesis Kit (New England Biolabs), as recommended by manufacturer's instructions. All RTqPCR reactions were performed in triplicate using TaqMan probes in combination with CFX96 Touch Real-Time PCR Detection System (BioRad), paired with CFX Maestro Analysis Software.

Each reaction mixture (20 μl) reaction mixture containing 1 μl cDNA template (100ng/ml), 1 μl FAM probe and 10 μl TaqMan Universal Master Mix (Applied biosystems by Thermo Fisher Scientific, Epsom, UK) was amplified as follows: denaturation at 95˚C for 10 minutes and 39 cycles at 95˚C for15 seconds, 60˚C for 30 seconds. Four hypoxia-inducible factor dependent genes ($egln3$: Dr03095294_m1, $pfkfb3$: $Dr03133482\_m1$, $vegfab$: Dr03072613_m1 $and$ $slc2a1a$: $Dr03103605\_m1$) and four glucocorticoid dependent genes ($fkbp5$: $Dr03114487\_m1$, $il6st$: Dr03431389_m1, $pck1$: $Dr03152525\_m1$ $and$ $lipca$: Dr03113728_m1) were quantified in the present study (Applied biosystems by Thermo Fisher Scientific, Epsom, UK).

Expression levels for each gene were normalized to e$ef1a1$ (Dr03432748_m1) and/or $rps29$ (Dr03152131_m1) and fold change values were generated relative to wild-type DMSO treated control levels, according to ΔΔCT method [82]. All data were expressed as fold change mean ± s.e.m and P ≤ 0.05 was considered statistically significant.

## Quantifying *phd3*:*eGFP*-related brightness

Images were acquired using Leica Application Suite version 4.9, which allowed the capture both of bright-field and GFP fluorescent images. To quantify the *phd3*:*eGFP*-related brightness of live embryos derived from each incrossed mutant line used in this project, Fiji (Image J) software v.2.0.0 was used. Images were converted into a grey scale 8-bit format and subsequently analysed by the software, by summing the grey values of all the pixels in the selected area, divided by the number of pixels. By default, since values equal to 0 are assigned to black and values equal to 255 to white, the quantified mean grey values are proportional to the intensity of the eGFP-related brightness expressed in the embryos. In particular, head, liver and tail (from the anus to the caudal peduncle) related brightness were selected and measured in all the mutant lines used in this study (**S1D Fig**). Genotyping post phenotypic analysis on *phd3*: *eGFP* sorted larvae confirmed the genotype-phenotype correlation.

## Statistical analysis

GraphPad Prism version 8.0 for MacOS (GraphPad Software, La Jolla, California, USA, www. graphpad.com) was used to perform statistical analysis on all the samples analysed. Unpaired t tests were used to test for significant differences between two sample groups (i.e cortisol quantification). One-way ANOVA was used for assessing mean grey values data quantification, whereas two-way ANOVA was used to evaluate qPCR data. As post-hoc correction tests, Sidak's method for multiple comparisons was used on normally distributed populations following one-way ANOVA, while Dunnett's correction was used for comparing every mean to a control mean, on normally distributed populations following two-way ANOVA.

## Supporting information

**S1 Fig. *arnt1*$^{-/-}$; *vhl*$^{-/-}$ larvae showed a reduced *phd3*:*eGFP* brightness and a partially rescued *vhl* phenotype.** A. Statistical analysis performed on mean gray value quantification (at the level of the head, liver and tail), after phenotypic analysis on 5dpf DMSO and BME [30μM] treated *arnt1*$^{+/-}$;*vhl*$^{+/-}$(*phd3*:*eGFP*) x *arnt1*$^{-/-}$; *vhl*$^{+/-}$(*phd3*:*eGFP*) derived larvae (n = 540). *vhl*$^{-/-}$ DMSO treated n = 17 larvae: head 166.67 ± 9.63 (mean ± s.e.m); liver 138.61 ± 12.05 (mean ± s.e.m); tail 50.31 ± 4.51 (mean ± s.e.m). *arnt1*$^{-/-}$;*vhl*$^{-/-}$ DMSO treated n = 13 larvae: head 121.05 ± 6.99 (mean ± s.e.m); liver 49.61 ± 3.88 (mean ± s.e.m); tail 21.75 ± 1.12 (mean ± s.e.m). *vhl*$^{-/-}$ BME treated n = 18 larvae: head 199.88 ± 7.71 (mean ± s.e.m); liver 222.57 ± 8.72 (mean ± s.e.m); tail 57.57 ± 4.11 (mean ± s.e.m). *arnt1*$^{-/-}$;*vhl*$^{-/-}$ BME treated n = 12 larvae: head 153.71 ± 8.66 (mean ± s.e.m); liver 62.58 ± 5.16 (mean ± s.e.m); tail 25.82 ± 1.54 (mean ± s.e.m). Ordinary One-way ANOVA followed by Sidak's multiple comparison test (*P < 0.05; **P < 0.01; ***P <0.001; ****P < 0.0001). B. Kaplan-Meier survival curves of the zebrafish *arnt1*$^{+/-}$; *vhl*$^{+/-}$(*phd3*:*eGFP*) genotype analysed in this study. Time is shown in days. Siblings n = 30; *arnt1*$^{-/-}$; *vhl*$^{-/-}$(*phd3*:*eGFP*) n = 8. The Log-rank (Mantel-Cox) test was used for statistical analysis. *arnt1*$^{-/-}$; *vhl*$^{-/-}$(*phd3*:*eGFP*) vs. siblings: **P < 0.0027. C. RTqPCR analysis performed on *arnt1* siblings (n = 10; 3 repeats) and *arnt1*$^{-/-}$ (n = 10; 3 repeats) larvae at 5 dpf, using *egln3* as probe. Statistical analysis was performed on ΔΔCt values, whereas data are shown as fold change values. Ordinary Two-way ANOVA followed by Dunnett's multiple comparison test (***P <0.001;****P < 0.0001). D. Representative picture of head, liver and tail areas selected in each larva to quantify the phd3:eGFP-related brightness via mean grey value quantification (Fiji, ImageJ software).
(TIF)

**S2 Fig. GC target genes expression in the presence of high, moderately upregulated and suppressed HIF signalling pathway.** Schematic view of RTqPCR analysis on *il6st*, *pck1* and *lipca* (GC target genes) expression performed on the following mutant lines: *vhl*$^{+/-}$*(phd3: eGFP)*, *arnt1*$^{+/-}$*;vhl*$^{+/-}$*(phd3:eGFP)* and *arnt1*$^{+/-}$*(phd3:eGFP)*. Statistical analysis performed on ΔΔCt values; data are shown as fold change values for RTqPCR analysed samples; ordinary Two-way ANOVA followed by Dunnett's multiple comparison test (*P < 0.05; **P < 0.01; ***P <0.001; ****P < 0.0001).
(TIF)

**S3 Fig. *gr*$^{-/-}$; *vhl*$^{-/-}$ larvae showed a reduced *phd3:eGFP* brightness and a partially rescued Vhl phenotype.** A-D'. Representative pictures of WISH performed on DMSO and BME [30 μM] treated arnt1 mutant line, at 5 dpf, using *cyp17a2* as probe. A-A') *arnt1* wt DMSO treated larvae (n = 26/28) showed normal *cyp17a2* expression, whereas 2/28 larvae showed a weaker one; B-B') *arnt1* wt BME treated larvae (n = 28/30) showed downregulated *cyp17a2* expression, whereas 2/30 larvae showed a normal one. C-C') In contrast, *arnt1*$^{-/-}$ DMSO treated larvae (n = 24/28) showed upregulated *cyp17a2* expression, whereas 4/28 larvae showed a weaker one. D-D') *arnt1*$^{-/-}$ BME treated larvae (n = 25/29) showed downregulated *cyp17a2* expression, whereas 4/29, showed a normal one. Chi-square test (****P < 0.0001). Scale bar 200 μm. E-H'. Representative pictures of WISH performed on DMSO and BME [30 μM] treated *vhl* mutant line, at 5 dpf, using *cyp17a2* as probe. E-E') DMSO treated *vhl* siblings (n = 18/21) showed normal *cyp17a2* expression, whereas 3/21 larvae showed a weaker one; F-F') BME treated *vhl* siblings (n = 28/30) showed downregulated *cyp17a2* expression, whereas 2/30 larvae showed a normal one. G-G') On the other hand, *vhl*$^{-/-}$ DMSO treated larvae (n = 27/28) showed weak *cyp17a2* expression, whereas 1/28 larvae showed a normal *one*. H-H') *vhl*$^{-/-}$ BME treated larvae (n = 30/30) showed downregulated *cyp17a2* expression. Chi-square test (****P < 0.0001). Scale bar 200 μm. I-I'. Representative picture of the colour threshold area calculation method (ImageJ software's tool) used to quantify the area occupied by the *cyp17a2* WISH staining both in *arnt1* siblings (n = 9) and *arnt1*$^{-/-}$ (n = 9). I'. unpaired t-test (****P <0.0001).
(TIF)

**S4 Fig. *gr*$^{-/-}$; *vhl*$^{-/-}$ larvae showed a reduced *phd3:eGFP* brightness and a partially rescued *vhl* phenotype.** A. Statistical analysis performed on mean gray value quantification (at the level of the head, liver and tail), after phenotypic analysis on 5dpf DMSO and BME [30μM] treated *gr*$^{+/-}$*;vhl*$^{+/-}$*(phd3:eGFP)* x *gr*$^{-/-}$*; vhl*$^{+/-}$*(phd3:eGFP)* derived larvae (n = 600). *vhl*$^{-/-}$ DMSO treated n = 9 larvae: head 186 ± 15.12 (mean ± s.e.m); liver 177.01 ± 20.85 (mean ± s.e.m); tail 62.34 ± 7.27 (mean ± s.e.m). *gr*$^{-/-}$*;vhl*$^{-/-}$ DMSO treated n = 7 larvae: head 106.96 ± 3.21 (mean ± s.e.m); liver 60.75 ± 2.56 (mean ± s.e.m); tail 30.67 ± 1.27 (mean ± s.e.m). *vhl*$^{-/-}$ BME treated n = 14 larvae: head 224.32 ± 6.83 (mean ± s.e.m); liver 244.07 ± 5.31 (mean ± s.e.m); tail 80.51 ± 5.49 (mean ± s.e.m). *gr*$^{-/-}$*;vhl*$^{-/-}$ BME treated n = 9 larvae: head 125.85 ± 3.6 (mean ± s.e.m); liver 63.56 ± 2.91 (mean ± s.e.m); tail 33.67 ± 1.02 (mean ± s.e.m). Ordinary One-way ANOVA followed by Sidak's multiple comparison test (*P < 0.05; **P < 0.01; ***P <0.001; ****P < 0.0001). B. Kaplan-Meier survival curves of the zebrafish *gr*$^{+/-}$; *vhl*$^{+/-}$*(phd3: eGFP)* genotype analysed in this study. Time is shown in days. Wild-types n = 20; *gr*$^{+/-}$; *vhl*$^{+/-}$ n = 20; *gr*$^{-/-}$; *vhl*$^{-/-}$*(phd3:eGFP)* n = 5. The Log-rank (Mantel-Cox) test was used for statistical analysis. *gr*$^{-/-}$; *vhl*$^{-/-}$*(phd3:eGFP)* vs. *gr*$^{+/-}$; *vhl*$^{+/-}$, ****P < 0.0001; *gr*$^{-/-}$; *vhl*$^{-/-}$*(phd3:eGFP)* vs. *wt*, ****P < 0.0001.
(TIF)

**S5 Fig.** *gr*$^{-/-}$;*arnt1*$^{-/-}$;*vhl*$^{-/-}$ **showed an even more reduced** *phd3:eGFP* **brightness.** Statistical analysis performed on mean gray values quantification (at the level of the head, liver and tail), after phenotypic analysis on 5dpf *gr*$^{+/-}$;*arnt1*$^{+/-}$*vhl*$^{+/-}$(*phd3:eGFP*) incross-derived GFP$^{+}$ larvae (n = 488). *vhl*$^{-/-}$ n = 5 larvae: head 125.82 ± 13.05 (mean ± s.e.m); liver 98.52 ± 3.8 (mean ± s.e.m); tail 37.43 ± 2.45 (mean ± s.e.m). *gr*$^{-/-}$;*arnt1*$^{-/-}$;*vhl*$^{-/-}$ n = 5 larvae: head 40.24 ± 2.46 (mean ± s.e.m); liver 26.07 ± 1.31 (mean ± s.e.m); tail 11.22 ± 0.47 (mean ± s.e.m); unpaired t-test (***P = 0.0002; ****P < 0.0001).
(TIF)

**S6 Fig. CRISPR/Cas9 injection per se does not affect HIF signalling.** A-D. Representative pictures of 5 dpf CRISPANT mutants created by redundantly targeting *lamb1b* gene via co-injection of 4x gRNAs in *vhl*$^{+/-}$(*phd3:eGFP*) incross-derived embryos (n = 400). Uninjected embryos were used as control (n = 470). Fluorescence, exposure = 991,4 ms. Scale bar 500 μm. E. Statistical analysis performed on mean grey values quantification (at the level of the head, liver and tail), after phenotypic analysis on 5 dpf *lamb1b* 4x gRNAs injected and uninjected *vhl*$^{+/-}$(*phd3:eGFP*) incross-derived larvae. *vhl*$^{-/-}$ uninjected n = 24 larvae: head 54.83 ± 3.68 (mean ± s.e.m); liver 77.86 ± 6.46 (mean ± s.e.m); tail 19.56 ± 1.43 (mean ± s.e.m). *vhl*$^{-/-}$ injected n = 25 larvae: head 59.74 ± 4.05 (mean ± s.e.m); liver 83.23 ± 5.92 (mean ± s.e.m); tail 19.9 ± 1.38 (mean ± s.e.m); unpaired t-test (all panels: *P < 0.05; **P < 0.01; ***P <0.001; ****P < 0.0001).
(TIF)

## Acknowledgments

We thank the University of Sheffield aquarium staff for the excellent care of fish stocks. We are grateful to Rosemary Kim, Helen Eachus, Emily Noël, Chris Derrick, Jack Paveley and Dheemanth Subramanya for useful discussions contributing to this study and for sharing chemical compounds. We finally thank Elisabeth Kugler for her technical support for image analysis.

## Author Contributions

**Conceptualization:** Davide Marchi, Fredericus J. M. van Eeden.

**Data curation:** Davide Marchi, Fredericus J. M. van Eeden.

**Formal analysis:** Davide Marchi.

**Funding acquisition:** Davide Marchi, Nils Krone, Fredericus J. M. van Eeden.

**Investigation:** Davide Marchi.

**Methodology:** Davide Marchi, Fredericus J. M. van Eeden.

**Project administration:** Davide Marchi, Fredericus J. M. van Eeden.

**Resources:** Kirankumar Santhakumar, Eleanor Markham, Nan Li, Karl-Heinz Storbeck, Nils Krone, Vincent T. Cunliffe, Fredericus J. M. van Eeden.

**Software:** Davide Marchi, Karl-Heinz Storbeck, Fredericus J. M. van Eeden.

**Supervision:** Davide Marchi, Fredericus J. M. van Eeden.

**Validation:** Davide Marchi, Nan Li, Vincent T. Cunliffe, Fredericus J. M. van Eeden.

**Visualization:** Davide Marchi, Fredericus J. M. van Eeden.

**Writing – original draft:** Davide Marchi, Fredericus J. M. van Eeden.

**Writing – review & editing:** Davide Marchi, Kirankumar Santhakumar, Eleanor Markham, Nan Li, Karl-Heinz Storbeck, Nils Krone, Vincent T. Cunliffe, Fredericus J. M. van Eeden.

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
