## [Decision Letter · Decision Letter 0]

9 Dec 2019

Dear Dr Marchi,

Thank you very much for submitting your Research Article entitled 'Bidirectional crosstalk between Hypoxia-Inducible Factor and glucocorticoid signalling in zebrafish larvae.' to PLOS Genetics. Your manuscript was fully evaluated at the editorial level and by independent peer reviewers. The reviewers appreciated the attention to an important problem, but raised some substantial concerns about the current manuscript. Based on the reviews, we will not be able to accept this version of the manuscript, but we would be willing to review again a much-revised version. We cannot, of course, promise publication at that time.

If you decide to revise the manuscript for further consideration at PLOS Genetics, please aim to resubmit within the next 60 days, unless it will take extra time to address the concerns of the reviewers, in which case we would appreciate an expected resubmission date by email to plosgenetics@plos.org.

[LINK]

We are sorry that we cannot be more positive about your manuscript at this stage. Please do not hesitate to contact us if you have any concerns or questions.

Yours sincerely,

Daniel A. Gorelick, Ph.D.

Guest Editor

PLOS Genetics

Gregory P. Copenhaver

Editor-in-Chief

PLOS Genetics

The reviewers were enthusiastic about the interesting results presented in your manuscript, showing that HIF signaling regulates responses to glucocorticoids. However, all reviewers were concerned about the readability and organization of the manuscript, making it difficult to appreciate the high quality results. A revised manuscript should pay particular attention to improving the organization and the writing to maximize clarity, in addition to addressing the specific reviewer comments below.

Reviewer's Responses to Questions

**Comments to the Authors:**

Reviewer #1: See attachment

Reviewer #2: The manuscript by Marchi et al, “Bidirectional crosstalk between Hypoxia-Inducible Factor and glucocorticoid signaling in zebrafish larvae” provides new information about the crosstalk of hypoxia and glucocorticoid signaling in vivo.

In addition to the scientific conclusions presented, the authors have produced another GR (nr3c1) mutant allele (currently 9 listed on zfin.org) and a new HIF1B (arnt) mutant (5 listed on ZFIN). Many of the nr3c1 alleles are similar to the new allele, but potentially published after initiation of this work. The arnt alleles represent mass deposited alleles from Sanger or the NIH, which likely were not studied, and are therefore not certain loss of function or nulls. My primary concern is that the organization and presentation of the data in the current version of this manuscript is cumbersome. The figures have multiple panels that when printed are not legible as the fonts are far too small. In some cases they are even hard to see on my monitor zoomed to greater than 200%. This made the key elements of this paper difficult to readily grasp. However, the key findings are important. With a significant overhaul of presentation, including perhaps rethinking figure organization would greatly improve both the impact and reach of the science presented by the authors. Below I have listed some additional major/minor concerns.

Other Major concerns:

QRT data shows mixed results (by gene target) which are not brought into the model or discussion. Why is BME, a GR agonist not able to increase expression of GR target genes? For some of these targets the mutant background appears to change basal expression levels and the BME still increases expression but not to the same degree as in wild type. Is the ratio of the increase from baseline to BME rates similar? This is difficult to see in figures.

Figure 2F and genotyping of mutants. The methods do not detail how mutants were genotyped. To do the cortisol assays in 2F the genotype of the larvae would have to be known prior to collection as a group for the ELISA. How was this done, is this indeed basal cortisol or acute stress, etc. (Minor) why is this expressed per 150 embryos on the y-axis?

Figure 3B image is poor, data is not new per se so could be mentioned (not shown) or moved to supplemental to make room for key novel data. Fig 3B’ these images do not readily demonstrate changes in expression.

Loss of MR in crispant appears to more severely impact HIF-reporter line than GR. How well do the guides target MR in Crispants (no molecular data is shown for Crispant efficiency)? How does the model reflect or change with the possibility that MR is potentially a major mediator of HIF:GC interactions, rather than GR? This is a really interesting observation that would be better studied in a MR mutant background however, I do not think that is a reasonable ask your manuscript has provided a lot of great data. However, discussing this outcome and possible meaning for future research is necessary.

Minor comments:

The authors use a HIF-reporter line, eGFP driven by the phd3 promoter, and a previously published Von Hippel Lindau (VHL) mutant that increases HIF function and activates EGFP in the HIF-reporter line. When scoring GFP changes they quantify within the head, liver, and tail. A single image showing how these regions were scored would be helpful.

In both Figure 1 and Figure 2 the gene map is presented in reverse (as if looking at a whole genome map). However, there is no reason not to orient the single genes from 5’ to 3’ in these figures. For Figure 2 the arrow that indicates the mutation site is pointing at exon 4 and not exon 3.

There could be more subpanel designators in figures. For example, Figure 1 has insets without designators, and this makes it more difficult to find what the author is referencing in the text.

There are some locations in the text and figures (including supplemental) that use hif1ßsh544, arnt1sh544 and at least one artn1sh544.

In Figure 2A and FigS2 the Y-axes range is large due to BME treatment and thus it becomes difficult to see physiological response in the absence of GR agonist.

In situ hybridization is difficult to get quantitative data from, though qualitatively as shown it is observed to be higher or lower expression. Could QRT of the same target genes (pomc and cyp17a2) provide better quantification?

Figure 3C. No direct comparison to vhl-/-, which is in other locations, but comparing GFP that is not side by side is not always easy. Overall, I’m wondering if GFP quantification data should be primary and images backing that up (sometimes in supplemental). Would this help create room in figures to help with legible presentation?

Figure 4. Were all images taken then genotyped? Seems that levels of GFP would be difficult to simply sort and verify. These details are not in methods.

Figure 6. MR was shown to not have a role in rapid non-genomic behaviors that required HPA axis signaling in zebrafish (Lee et al, Genes Brain Behavior 2018). This is contrary to Faught & Vijayan manuscript published at the same time and referenced herein.

The clarity of the paper will depend on clear definitions and use of terms. For example, GC response can be a lot of things, but I think in the discussion model it refers to GR transcriptional regulation. Suppressed HIF/upregulated HIF is generally tied to the HIF-reporter, so again is a transcriptional activity. In the models in Fig 2, there is no “cortisol” step as in the discussion model, so GC response could mean cortisol release there and maybe does? This should be clear and word choice around glucocorticoids (e.g. cortisol) and GR and GR transcriptional activity should be clear.

The final model should be worked up a little bit to not only show what you think is happening but perhaps point to supporting data in figures (again figures need better subpanels to make that easier).

The study uses Betamethasone, a glucocorticoid receptor (GR) agonist that has been used clinically for decades and your group recently (2017) showed can activate HIF. Although this GR agonist primarily interacts with GR there is some minimal crosstalk with mineralocorticoid receptor (MR), but this small crosstalk may have biological relevance.

Reviewer #3: This manuscript looks at the interaction between glucocorticoids and the HIF pathway. It generates and employs some useful genetic mutants and presents an important addition to the growing field of integration of different stress pathways on physiology. At times it is rather convoluted and complicated, and lacks clear logical explanations at parts. If the authors can present the conclusions in a clearer model, the paper would be much improved.

The authors use the term “attenuated” to describe the partially elevated HIF signaling in the vhl;arnt1 double mutants, which is a little confusing as it is only attenuated with respect to vhl single mutants and not with respect to the WT groundstate. I know what the authors mean, but I suggest they replace this throughout with the term “moderately upregulated” or similar for clarity.

For the statement that “In contrast, when the HIF pathway was suppressed (arnt1-/-), BME

was able to further upregulate mainly the expression of fkbp5” the authors need to show the fold upregulation and associated ANOVA Post-test comparing the upregulation of fkbp5, il6st and pck1 in the arnt1 mutants with and without BME treatment as well as with sibs treated with BME.

Figure 2B’ is confusing as it implies mutation of arnt1 inhibits suppression of Hif. This pathway needs to be presented in a more standard manner ie

Hif--| GCGR--| pomca

Similarly the Figure 2C' complicates matters. I appreciate that in the case of increased Hif signalling, a different mechanism is being employed but this seems to contradict Fig2B’. Could the two be combined into one clear model?

Upregulation of cyp17a2 in arnt1 mutants is claimed, but not decisively shown. The insitu image does not convincingly demonstrate upregulation

Figure 3C’ is too small. I cannot see it properly but it appears that arnt1-/-;vhl-/- double mutants have more egln3, and vegfab than the vhl-/- mutants alone. Is that correct on the graph? That seems odd and contrary to my understanding of the hif pathway logic.

The CRISPANT approach was developed by Burger et al 2016, not Wu et al 2018. The former should be cited rather than the latter.

Given that the authors have previously shown that GR activates the Hif pathway by degrading VHL (Vettori et al 2017), how do they explain the rescue of vhl mutants by GR mutation, which should be hypostatic to vhl.

“Consequently, we speculate that in an arnt1 knock-out scenario, pomca downregulation occurs as a consequence of glucocorticoid induced negative feedback loop (aimed to shutdown cortisol biosynthesis). In contrast, as upregulated HIF levels appear to repress pomca expression, we speculate that this occurs in a way that resembles the GR-mediated negative feedback loop.” – I cannot make sense of this statement and needs to be written with much more clarity. The latter sentence invokes the same feedback loop as for arnt1 knockout.

**Have all data underlying the figures and results presented in the manuscript been provided?**

Reviewer #1: Yes

Reviewer #2: Yes

Reviewer #3: Yes

PLOS authors have the option to publish the peer review history of their article (what does this mean?). If published, this will include your full peer review and any attached files.

Reviewer #1: No

Reviewer #2: Yes: Karl J. Clark

Reviewer #3: No

---

## [Decision Letter · Decision Letter 1]

20 Mar 2020

Dear Dr Marchi,

Thank you very much for submitting your Research Article entitled 'Bidirectional crosstalk between Hypoxia-Inducible Factor and glucocorticoid signalling in zebrafish larvae.' to PLOS Genetics. Your manuscript was fully evaluated at the editorial level and by independent peer reviewers. The reviewers appreciated the attention to an important topic but identified some aspects of the manuscript that should be improved.

We therefore ask you to modify the manuscript according to the review recommendations before we can consider your manuscript for acceptance. Your revisions should address the specific points made by each reviewer.

[LINK]

Yours sincerely,

Daniel A. Gorelick, Ph.D.

Guest Editor

PLOS Genetics

Gregory P. Copenhaver

Editor-in-Chief

PLOS Genetics

Thank you for responding to the reviewer's comments. The revised manuscript is substantially improved. We feel that the revised manuscript is potentially suitable for publication, pending your response to the reviewer's comments. Please pay particular attention to comments regarding Figure 2, in which there are different results in the revised manuscript versus the original submission. We look forward to seeing a revised version of your manuscript.

Reviewer's Responses to Questions

**Comments to the Authors:**

Reviewer #1: The authors have addressed most of the issues that I raised and have significantly improved the manuscript. However, I still have a few concerns, which mainly concern the data presented in Figure 2, the description of these results and the part of the Discussion that deals with these results. Most importantly, I think the authors should be more careful with their interpretation of the data, which is at times speculative.

- To my big surprise, the data in Fig.2A’ are different from the results that were originally presented for this experiment. Why were these data substituted? And why were the data from the double mutant removed?

- The authors use the qPCR data to prove that HIF signaling induces a glucocorticoid-resistant state. However, I do not think these data are entirely convincing, because the glucocorticoid response in the siblings is highly variable, as well as the basal expression levels in the mutant (why for example the fold change upon BME treatment in the (allegedly glucocorticoid resistant) vhl mutant is higher than in wild type). The authors should at least be more careful with statements about the glucocorticoid resistant state since it is solely based on these data.

- In Fig.2G, the arrow from low pomca to high cortisol in the arnt1 mutant is still difficult to explain, although the increased GC/GR response could play a role, as the authors suggest. They should be a bit more carefl here as well, and at least make clear that the combination high cortisol/low pomca is very rare and difficult to explain (and that this combination may change over the course of development).

- The altered response to BME should be incorporated in the top panel of Fig.2G, by drawing a direct arrow from HIF to GC/GR response (with BME next to that arrow). Now it looks like the increased response to BME is a result of HIF acting through pomca and cortisol.

-

Minor points:

- Page 6, line 18: Hif-1β is not a nuclear receptor.

- Page 7, line 5: please mention that the levels are still higher than the wild type, and preferably also why these levels are not shown in Fig.1K (I assume because they are undetectable?).

- In the figures, sometime controls are labeled ‘sibs’ and sometimes ‘wt’. If there is no particular reason for this difference, please make it consistent.

Reviewer #2: The revised manuscript by March et al has been vastly improved by the responses of the authors to the reviewers. There remain some minor typos. Example from abstract "...both the glucocorticoid receptor (Gr) responsiveness and the endogenous cortisol..." could be "...both glucocorticoid receptor (Gr) responsiveness and endogenous cortisol...". Also, in reference to Gr and Mr the authors state "Together they act as a transcription factor, which can function in either a genomic or in non-genomic way". This is confusing because in their role as a transcription factor that is by definition a genomic response by changing transcription. However both, particularly GR, are likely involved in other cell signaling that does not involve it's conventional role as a transcription factor and this is referred to as non-genomic.

Overall, this paper which by its very nature can be complicated is vastly improved. Thank you for your efforts.

**Have all data underlying the figures and results presented in the manuscript been provided?**

Reviewer #1: Yes

Reviewer #2: Yes

PLOS authors have the option to publish the peer review history of their article (what does this mean?). If published, this will include your full peer review and any attached files.

Reviewer #1: No

Reviewer #2: Yes: Karl J. Clark

---

## [Editor Report · Decision Letter 2]

3 Apr 2020

Dear Dr Marchi,

We are pleased to inform you that your manuscript entitled "Bidirectional crosstalk between Hypoxia-Inducible Factor and glucocorticoid signalling in zebrafish larvae." has been editorially accepted for publication in PLOS Genetics. Congratulations!

Yours sincerely,

Daniel A. Gorelick, Ph.D.

Guest Editor

PLOS Genetics

Gregory P. Copenhaver

Editor-in-Chief

PLOS Genetics

Comments from the reviewers (if applicable):

**Data Deposition**

http://datadryad.org/submit?journalID=pgenetics&manu=PGENETICS-D-19-01743R2

**Press Queries**

---

## [Editor Report · Acceptance letter]

29 Apr 2020

PGENETICS-D-19-01743R2 

Bidirectional crosstalk between Hypoxia-Inducible Factor and glucocorticoid signalling in zebrafish larvae 

Dear Dr Marchi, 

We are pleased to inform you that your manuscript entitled "Bidirectional crosstalk between Hypoxia-Inducible Factor and glucocorticoid signalling in zebrafish larvae" has been formally accepted for publication in PLOS Genetics! Your manuscript is now with our production department and you will be notified of the publication date in due course.

With kind regards,

Matt Lyles

PLOS Genetics

On behalf of:
